# Proteostasis is differentially modulated by inhibition of translation initiation or elongation

Khalyd J Clay, Yongzhi Yang, Christina Clark, Michael Petrascheck*

Department of Molecular Medicine, Department of Neuroscience, Scripps Research Institute, La Jolla, United States

**Abstract** Recent work has revealed an increasingly important role for mRNA translation in maintaining proteostasis. Here, we use chemical inhibitors targeting discrete steps of translation to compare how lowering the concentration of all or only translation initiation-dependent proteins rescues *Caenorhabditis elegans* from proteotoxic stress. We systematically challenge proteostasis and show that pharmacologically inhibiting translation initiation or elongation elicits a distinct protective profile. Inhibiting elongation protects from heat and proteasome dysfunction independently from HSF-1 but does not protect from age-associated protein aggregation. Conversely, inhibition of initiation protects from heat and age-associated protein aggregation and increases lifespan, dependent on *hsf-1*, but does not protect from proteotoxicity caused by proteasome dysfunction. Surprisingly, we find that the ability of the translation initiation machinery to control the concentration of newly synthesized proteins depends on HSF-1. Inhibition of translation initiation in wild-type animals reduces the concentration of newly synthesized proteins but increases it in *hsf-1* mutants. Our findings suggest that the HSF-1 pathway is not only a downstream target of translation but also directly cooperates with the translation initiation machinery to control the concentration of newly synthesized proteins to restore proteostasis.

*For correspondence:
pscheck@scripps.edu

## Editor's evaluation

Inhibition of translation has been found as a conserved intervention to extend lifespan across a number of species. In this work, the authors systematically investigate the similarities and differences from pharmacological inhibition of protein synthesis at the initiation or elongation steps on longevity and stress resistance. These experiments are important for conceptualizing how translation inhibition actually extends lifespan and promotes proteostasis.

## Introduction

Protein synthesis is a highly regulated process involving the precise orchestration of many chaperones, co-factors, enzymes, and biomolecular building blocks. It is critical at every level of the life cycle, from development through aging, and is central to stress adaptation. Protein synthesis largely determines the folding load on the proteostasis network, which regulates protein production, folding, trafficking, and degradation to maintain a functional proteome. The imbalance of the proteostasis network caused by age-associated stress or acute environmental insults leads to misfolding of proteins, accumulation of aggregates, and eventually to disease (*Balch et al., 2008*).

A substantial body of work has revealed that the protein synthesis machinery directly participates in protein folding or aggregation. For example, in mice, point mutations in specific tRNAs or components of the ribosomal quality control pathway can lead to protein aggregation and neurodegeneration

(*Vo et al., 2018*; *Yonashiro et al., 2016*; *Chu et al., 2009*; *Nollen et al., 2004*; *Nedialkova and Leidel, 2015*). Similarly, early RNAi screens in the nematode *Caenorhabditis elegans* identified several ribosomal subunits whose knockdown increased the aggregation of polyglutamine (PolyQ) proteins (*Nollen et al., 2004*). These findings highlight the importance of translation in protein misfolding.

However, subsequent studies reveal a more intricate role of translation in protein aggregation. Depending on how translation is modulated, it leads to either decreased or increased protein aggregation. For example, RNAi-mediated knockdown of translation initiation factors increases lifespan, improves proteostasis, and reduces protein aggregation in *C. elegans* (*Balch et al., 2008*; *Rogers et al., 2011*; *McQuary et al., 2016*; *Lan et al., 2019*; *Howard et al., 2016*). Similarly, work in yeast and cell culture shows that pharmacological inhibition of translation prior to a heat shock prevents proteins from aggregating (*Medicherla and Goldberg, 2008*; *Choe et al., 2016*; *Xu et al., 2016*; *Riback et al., 2017*). While studies in *C. elegans*, cell culture, and yeast agree that inhibition of translation reduces protein aggregation, their proposed underlying mechanisms differ. Overall, the proposed mechanisms can be categorized into two broad models on how lowering translation reduces protein aggregation.

The first model, referred to as the *selective translation model*, proposes that inhibition of translation is selective. In the selective translation model, the increased availability of ribosomes leads to differential translation of mRNAs coding for stress response factors and thus to increased folding capacity (*Rogers et al., 2011*; *McQuary et al., 2016*; *Lan et al., 2019*; *Seo et al., 2013*). In general, studies proposing a version of *the selective translation* model show that inhibiting translation requires HSF-1 to reduce protein aggregation (*Howard et al., 2016*; *Tye and Churchman, 2021*). In the *selective translation model*, protein aggregation is reduced by an HSF-1-dependent active generation of folding capacity to remodel the proteome.

The second model, referred to as the *reduced folding load model,* proposes that newly synthesized proteins are the primary aggregation-prone species of proteins. Therefore, newly synthesized proteins constitute the most significant folding load on the proteostasis machinery. Inhibition of translation reduces the concentration of newly synthesized proteins and, thus, the load on the folding machinery. In contrast to the *selective translation model*, which proposes selective protein synthesis of HSF-1-dependent stress response factors, the *reduced folding load model* does not depend on specific factors but generates folding capacity by reducing the overall folding load. A problem comparing previous studies has been their use of different model organisms, different proteostatic insults, and modes of inhibition. Furthermore, previous studies did not control how much protein synthesis was reduced by different modes of translation inhibition making direct comparisons between the *selective translation model* and the *reduced folding load model* difficult.

In this study, we systematically compared these two models in *C. elegans* using pharmacological agents to block various steps along the protein synthesis cycle but ensured to use inhibitors that achieve comparable reductions in the concentration of newly synthesized proteins across the different interventions. We characterize how lowering translation protects *C. elegans* from proteotoxic insults such as proteasome inactivation, heat shock, and aging. Our data reveal that the step inhibited in mRNA translation dictates which of the two protective mechanisms is activated. Furthermore, they uncover an unexpected link between translation initiation and HSF-1 and show that inhibition of translation initiation fails to lower the concentration of newly synthesized proteins in the absence of HSF-1.

## Results

### Characterizing mRNA translation inhibitors in *C. elegans*

We first set out to identify suitable pharmacological translation initiation (TI) and translation elongation (TE) inhibitors by screening a series of translation inhibitors for their ability to lower protein synthesis in *C. elegans* in a dose-dependent manner (*Dmitriev et al., 2020*). To monitor protein synthesis, we employed SUrface SEnsing of Translation (SUnSET) (*Arnold et al., 2014*). In this method, translating ribosomes incorporate puromycin into newly synthesized proteins. The level of puromycin incorporation serves as a quantitative measure for translation and is detected by western blotting using an anti-puromycin monoclonal antibody (*Figure 1A*). The six molecules examined all reduced puromycin incorporation relative to DMSO controls to varying degrees (*Figure 1B and C*). Based on their ability to lower the concentration of newly synthesized proteins by ~40–50% and their annotated

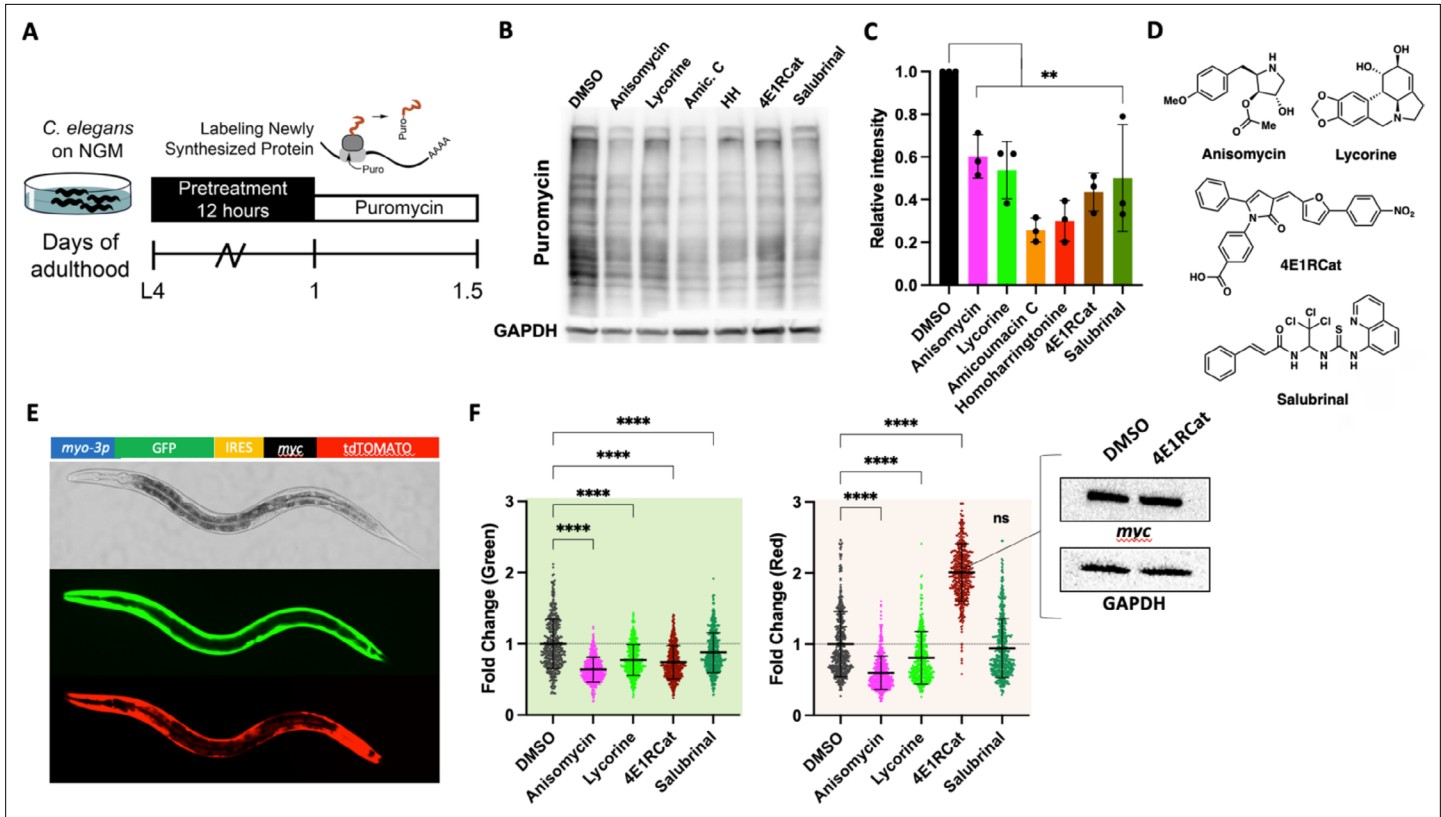

**Figure 1.** Identifying mechanistic inhibitors of protein synthesis in *C. elegans*. (**A**) Monitoring changes in protein synthesis using the SUrface SEnsing of Translation (SUnSET) method. *C. elegans* were treated with solvent (DMSO) or the indicated inhibitors (100 µM) for 12 hr, followed by a 4 hr puromycin incorporation. Worms were lysed, and protein extracts were run on SDS-PAGE gels, followed by staining with a puromycin-specific antibody. (**B**) Six translation inhibitors reduce puromycin incorporation relative to DMSO control. GAPDH was used as a loading control. (**C**) Quantification of three independent SUnSET experiments, as shown in (**B**). Significance was determined by one-way ANOVA with Dunnett's multiple comparisons tests where **=p ≤ 0.01 for all treatments. Error bars indicate mean ± SD from three independent trials. (**D**) Chemical structures of anisomycin, lycorine, 4E1RCat, and salubrinal. Note each is structurally distinct. (**E**) Representative pictures showing the expression pattern of the bi-cistronic reporter in L4 stage animals. The image in the brightfield channel shows an L4 stage animal. Images in green and red channels show that GFP and tdtomato are expressed in body wall muscle. (**F**) Fluorescence of 500 transgenic animals treated with anisomycin, lycorine, 4E1RCat, and salubrinal. Each treatment reduced the GFP signal (green shading), but only anisomycin and lycorine reduced the dtTomato signal (red shading). Significance was determined by one-way ANOVA with Dunnett's multiple comparisons tests where ****=p ≤ 0.0001 for all treatments. Error bars indicate mean ± SD. The experiment was repeated four times with similar results. Inset: 4E1RCat emits red fluorescence; therefore, the expression of tdTomato needed to be tested by western blot. 4E1Rcat does not change myc expression on the protein level, GAPDH is used as a loading control, a representative image of three independent experiments.

The online version of this article includes the following source data and figure supplement(s) for figure 1:

**Source data 1.** Unedited SUrface SEnsing of Translation (SUnSET) western blots.

**Source data 2.** Quantifications of western blots.

**Source data 3.** ChemDraw files for structures.

**Source data 4.** Summary of green/red integrated fluorescence and unedited myc/GAPDH blots.

**Figure supplement 1.** Cycloheximide, a ubiquitously used eukaryotic translation inhibitor, inhibits translation elongation (TE) in *C. elegans*.

**Figure supplement 1—source data 1.** Unedited SUrface SEnsing of Translation (SUnSET) western blots.

**Figure supplement 1—source data 2.** Quantifications of western blots.

**Figure supplement 1—source data 3.** Summary of green integrated fluorescence.

**Figure supplement 1—source data 4.** Summary of red integrated fluorescence.

targets (vide infra), we selected four structurally distinct inhibitors, anisomycin, lycorine, 4E1RCat, and salubrinal, to investigate further (*Figure 1D*).

The selected molecules have been extensively studied in yeast and cell culture to identify their purported mechanism of action—broadly, anisomycin and lycorine inhibit TE, while 4E1RCat and salubrinal inhibit TI. To confirm their action as TI or TE inhibitors in *C. elegans*, we generated a bi-cistronic reporter to co-express both GFP and tdTomato under the control of the *myo-3* promotor and an internal ribosomal entry site (IRES) (*Figure 1E*; *Li and Wang, 2012*). In transgenic worms carrying the *myo3p::*GFP-IRES-tdTomato construct, strong GFP and tdTomato fluorescence were observed primarily in the muscle tissue. Using this reporter, we measured the fluorescence of 500 L4 animals treated with one of the four translation inhibitors. As expected for the TE inhibitors, anisomycin and lycorine decreased fluorescence in both green and red channels.

On the other hand, the TI inhibitors 4E1RCat and salubrinal only decreased the fluorescence of the green channel, consistent with an initiation-dependent translation of the GFP mRNA sequence and the initiation-independent translation of the IRES-tdTomato sequence (*Figure 1F*). However, we observed a strong increase in red fluorescence in worms treated with 4E1RCat, but this is due to the red color of the molecule itself. We confirmed no change in tdTomato levels by western blot (*Figure 1F*, inset). The above results confirm that the four translation inhibitors function in distinct translation steps, consistent with previous reports in yeast and cell culture.

We also confirmed the mechanism of translation inhibition of cycloheximide. Although cycloheximide is a widely used inhibitor of translation throughout the literature, to our knowledge, it has not been shown to specifically act as a TE inhibitor in *C. elegans*. Using the SUnSET assay and our bi-cistronic reporter, we show that cycloheximide directly reduces the concentration of newly synthesized proteins (*Figure 1—figure supplement 1A, B*) and functions as a TE inhibitor since we observed a reduction of both the GFP and the tdTomato signal. Salubrinal, which we included as a TI inhibitor control, only reduced translation promoted by *myo-3* but not the IRES site as seen before (*Figure 1—figure supplement 1C, D*). We noted that the inhibition of translation by cycloheximide across different trials was considerably more variable compared to anisomycin and lycorine (compare *Figure 1B and C* with *Figure 1—figure supplement 1A, B*). We therefore decided to proceed with anisomycin and lycorine as TE inhibitors as they reliably reduced the concentration of newly synthesized proteins by 40–50%.

## Initiation and elongation inhibitors protect from thermal stress by HSF-1-dependent and -independent mechanisms, respectively

After confirming that the four translation inhibitors (*Figure 1D*) acted as TI or TE inhibitors in *C. elegans* as seen in mammals, we set out to investigate how pharmacologically targeting different steps of the translation cycle will affect stress-induced protein aggregation. We first chose thermal stress to induce protein aggregation and asked if both TI and TE inhibitors improve stress resistance. The animals were treated with either of the four inhibitors on day 1 of adulthood and, after a 72 hr treatment, moved to a non-permissive temperature of 36°C (*Figure 2A*). Hourly monitoring revealed that all four molecules significantly improved the survival of N2 animals (*Figure 2B*). These data show that both the inhibition of TI and TE protect from thermal stress.

We next asked if translation inhibitors require the canonical heat shock response (HSR) controlled by the transcription factor HSF-1 to protect from heat stress (HS). Previous work by us and others resulted in contradictory findings on whether protection from heat by translation inhibition depends on HSF-1 (*Seo et al., 2013*; *Solis et al., 2018*; *Zhou et al., 2014*). Therefore, to test if translation inhibition protects from HS in an HSF-1-dependent or -independent manner, we repeated the thermotolerance assay in HSR-deficient *hsf-1(sy441)* mutants. Only the TE inhibitors anisomycin and lycorine protected *hsf-1(sy441)* from HS-induced proteotoxicity, while the TI inhibitors 4E1RCat and salubrinal did not (*Figure 2C*). These results showed that different modes of translational inhibition protect by genetically separable mechanisms. However, this also reconciles previous contradictions as different groups inhibited translation using inhibitors or RNAi with specificity for either step.

To broadly confirm the ability of TE inhibitors to protect from thermal proteotoxicity through improved proteostasis, we conducted sequential detergent extractions to biochemically isolate and quantify soluble and insoluble proteins in wild-type (N2) animals (*David et al., 2010*; *Reis-Rodrigues et al., 2012*; *Simonsen et al., 2008*). Following a 2 hr HS at 36°C, we observed a substantial increase

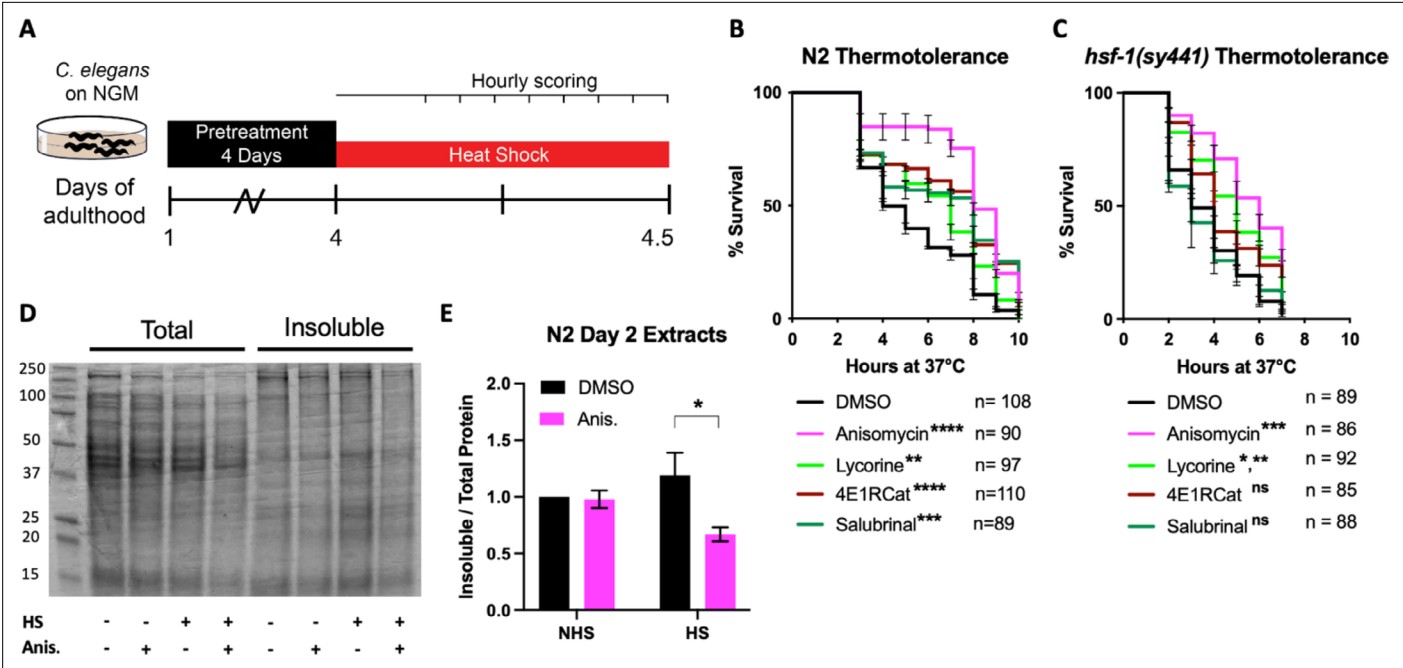

**Figure 2.** Initiation but not elongation inhibitors depend on HSF-1 to protect *C. elegans* from thermal stress. (**A**) Day 1 adult wild-type (**N2**) and *hsf-1(sy441)* animals were treated for 3 days, then transferred to NGM plates. They were then subjected to a constant, non-permissive temperature of 36°C (heat shock [HS]) and scored alive/dead every hour by movement. (**B**) Graph shows survival as a function of hours at 36°C of day 4 adult N2 animals pre-treated with 100 µM translation inhibitor. Data show the mean ± SEM from three independent trials where each measurement is at least: **=p ≤ 0.01, ***p≤0.001, and ****p≤0.0001 by row-matched two-way ANOVA with Šídák multiple comparisons test. (**C**) Graph shows survival as a function of hours at 36°C of day 4 adult *hsf-1(sy441)* animals pre-treated with 100 µM translation inhibitor. Data show the mean ± SEM from three independent trials where each measurement is at least: *=p ≤ 0.05, **=p ≤ 0.01, and ***p≤0.001 by row-matched two-way ANOVA with Šídák multiple comparisons test. (**D**) Representative SDS-PAGE gel stained with the protein stain Coomassie blue for visualization. Anisomycin (Anis.) reduces the proportion of detergent-insoluble protein following a 2 hr HS of N2 animals. Proteins were detergent extracted, ultracentrifuged, and the insoluble pellet was resuspended in 8 M urea before running on the gel. (**E**) Quantification of four separate extractions shows anisomycin significantly reduces HS-induced aggregation in wild-type N2 animals. Gels were stained with Sypro Ruby. Data are displayed as mean ± SEM and *=p ≤ 0.05 by two-tailed Student's t-test.

The online version of this article includes the following source data for figure 2:

**Source data 1.** Summary of thermotolerance survival numbers (**N2**).

**Source data 2.** Summary of thermotolerance survival numbers (*hsf-1(sy441)*).

**Source data 3.** Unedited SDS-PAGE gel.

**Source data 4.** Quantifications of insoluble gels.

in insoluble protein compared to non-heat-shocked controls. Furthermore, pre-treatment with the TE inhibitor anisomycin suppressed the increase in protein insolubility (*Figure 2D and E*), consistent with the observed thermal protection. These results reveal that TI inhibitors trigger an HSR-dependent mechanism to protect from thermal stress, while TE inhibitors trigger an HSR-independent mechanism. We concluded that different modes of translation inhibition protect the proteome by genetically distinct mechanisms.

## Translation elongation but not initiation inhibitors protect from heat shock-induced PolyQ aggregation

TE inhibitors are reported to free up folding capacity by reducing overall protein synthesis. To monitor folding capacity in live imaging, we developed an experimental procedure to examine how translation inhibition will dynamically affect protein aggregation in real time in the PolyQ::YFP strain AM140 (*Figure 3A*). The AM140 strain expresses a stretch of 35 glutamine residues fused to YFP (PolyQ::YFP) in the muscle, which we used as an orthogonal model of reduced protein folding capacity. Expression of the aggregation-prone PolyQ stretch increases the protein folding load on the proteostasis system. Thus, its aggregation propensity acts as a sensor for protein folding capacity (*Brignull et al., 2006*;

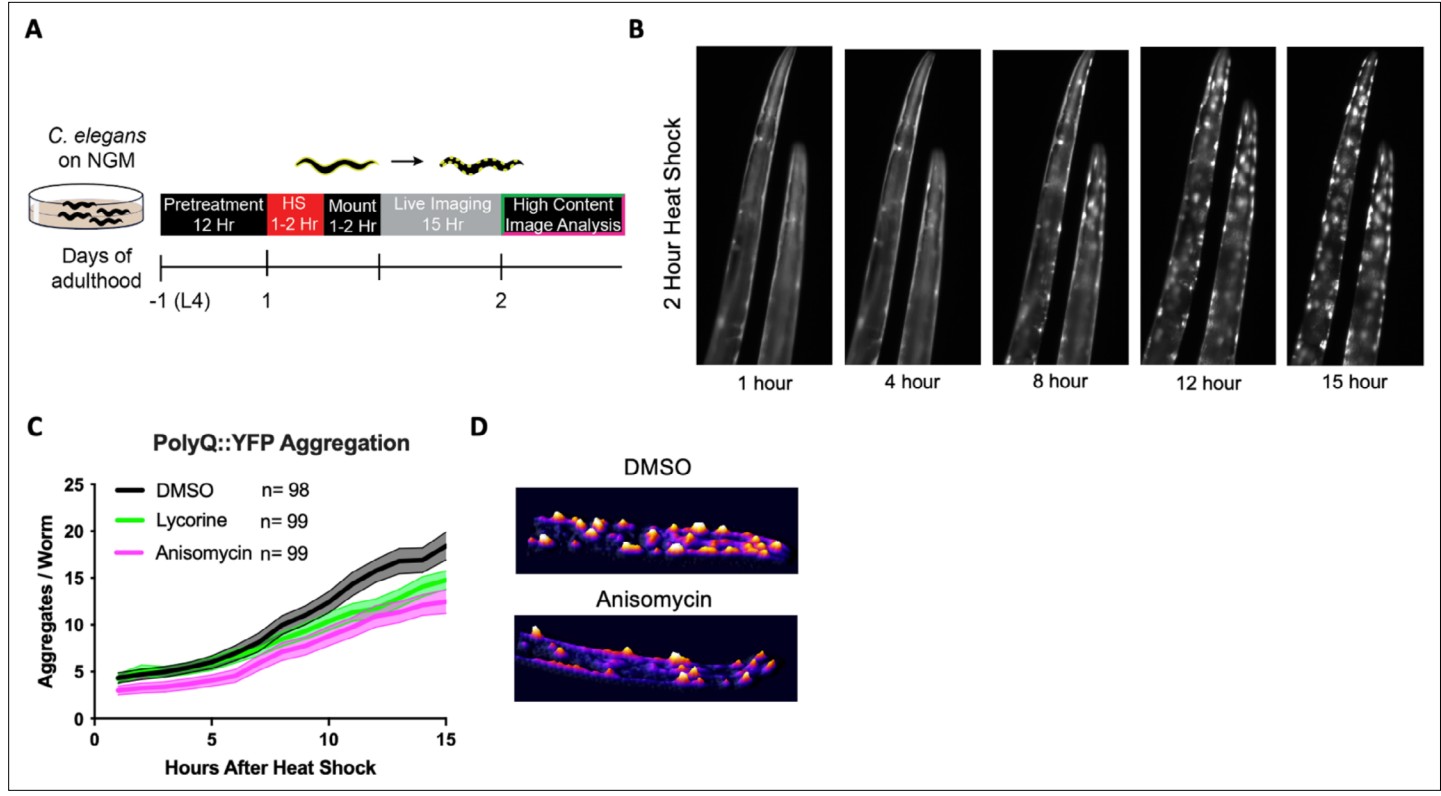

**Figure 3.** Elongation inhibitors reduce the number of heat shock-induced protein aggregates. (**A**) Day 1 AM140 adult worms expressing the polyglutamine-YFP fusion protein (PolyQ::YFP) in their muscle were subjected to heat stress (HS) on NGM plates for 2 hr at 36°C followed by a 1–2 hr mounting/immobilization procedure in 384-well plates and subsequent live imaging for 15 hr. (**B**) Fluorescent time-lapse images of two animals expressing the PolyQ::YFP fusion protein in the body wall muscle. The animals were embedded in the hydrogel for immobilization. Following a 2 hr HS, animals were imaged over 15 hr; by 8 hr, the YFP signal began to localize into discrete puncta that persisted through the observation time. (**C**) Graph shows the mean number of PolyQ aggregates per worm as a function of time following heat shock. *C. elegans* (PolyQ::YFP) were pre-treated with lycorine, anisomycin (100 µM), or DMSO. Lines indicate mean, and shading indicates 95% CI. (**D**) Representative images of control (top) and 100 µM anisomycin-treated (bottom) PolyQ animals 15 hr after HS. The representative images shown have been uniformly modified using the '3D Surface Plot' plugin in ImageJ to visualize aggregates.

The online version of this article includes the following source data and figure supplement(s) for figure 3:

**Source data 1.** Uncropped time-lapse fluorescence micrographs.

**Source data 2.** Summary of aggregation numbers.

**Source data 3.** Video of aggregation process with '3D Surface Plot' plugin.

**Figure supplement 1.** Requirements for protection from heat shock-induced polyglutamine (PolyQ) aggregation.

**Figure supplement 1—source data 1.** Quantification of heat shock time dependency.

**Figure supplement 1—source data 2.** Quantification of translation inhibitor screen HS-induced aggregation.

**Figure supplement 1—source data 3.** Quantification of time dependency for inhibitor treatment.

*Moronetti Mazzeo et al., 2012*). After a 2 hr HS, the initially diffuse PolyQ signal gradually localized into puncta (*Figure 3B*, *Video 1*). Aggregation foci formation resulted from a redistribution of the YFP signal into aggregation foci as the total level of YFP fluorescent signal for a given animal remained constant (*Figure 3—figure supplement 1A*).

Only TE inhibitors lycorine and anisomycin significantly reduced HS-induced PolyQ aggregation, while the TI inhibitors 4E1RCat had a small effect and salubrinal had no effect (*Figure 3C*, *Figure 3—figure supplement 1B*). Pre-treatment with anisomycin and lycorine reduced the number of PolyQ aggregates per worm. However, pre-treatment did not change the onset of aggregation, the rate of formation, or the final size of the aggregation foci (*Morley et al., 2002*; *Figure 3D*). Furthermore, we found the effect of anisomycin or lycorine to be time-dependent. They strongly inhibited aggregation following a 12 hr preincubation period but less so following a 4 hr preincubation (*Figure 3—figure*

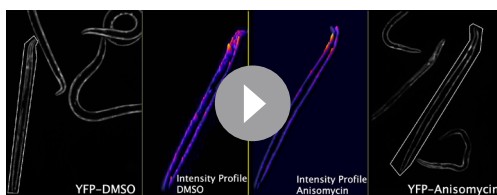

**Video 1.** Anisomycin prevents heat shock-induced polyglutamine (PolyQ) aggregation. AM130 *C. elegans* expressing 35 glutamine residues fused to YFP (PolyQ::YFP) were treated with DMSO or anisomycin for 12 hr before subjecting the animals to a 2 hr heat shock at 36°C. The animals were embedded into a hydrogel physically immobilizing the worm and imaged over 15 hr. The representative images shown have been uniformly modified using the '3D Surface Plot' plugin in ImageJ to visualize aggregates.

https://elifesciences.org/articles/76465/figures#video1

supplement 1C). Starting treatment with anisomycin after the HS did not reduce the number of aggregation foci (not shown). These results suggest that anisomycin and lycorine reduce the early formation of aggregate foci but do not alter the dynamics once aggregation begins.

## Translation elongation but not initiation inhibitors protect from proteasome dysfunction

The data thus far suggest that inhibition of translation protects from folding stress by an HSF-1-dependent and -independent mechanism determined by the inhibited translation step. The proteasomal system is another critical proteostasis mechanism by which cells clear protein aggregates. The proteasomal system ubiquitinylates misfolded proteins by ubiquitin ligases to target them for degradation by the 26S proteasome. Blocking proteasome degradation by bortezomib, a specific inhibitor of the 20S subunit, results in the formation of protein aggregates and proteotoxic stress (*Schrader et al., 2016*).

We pre-treated both N2 and *hsf-1(sy441)* L4 animals with either TI or TE inhibitors for 12 hr, followed by the addition of bortezomib, and measured survival on day 8 of adulthood (*Figure 4A*). We chose to include the *hsf-1(sy441)* background because it serves as a proxy of reduced folding capacity given the defective HSR, allowing us to further investigate the relationship between reduced translation and its conditional dependency on *hsf-1*. Compared to non-treated controls, bortezomib decreased the survival of both genotypes. TE inhibitors improved the survival of the *hsf-1(sy441)* animal back to untreated levels but only showed a non-significant tendency to improve the survival of N2 wild-type animals (*Figure 4B*). In contrast, TI inhibitors enhanced the proteotoxicity elicited by bortezomib in both backgrounds. We also observed bortezomib-treated *hsf-1(sy441)* animals to shrink in size (*sma* phenotype) (*Figure 4C*). We quantified this effect after a 12 hr pre-treatment of anisomycin and successive co-incubation of bortezomib at day 3 of adulthood. Anisomycin substantially rescued *sma* phenotypes, almost completely reversing the animals back to normal size (*Figure 4D*). A similar rescue was observed for lycorine but not quantified.

We next asked if the induction of the *sma* phenotype was specific to *hsf-1(sy441)* animals or if the combination of proteasome toxicity with impaired protein folding capacity caused it. We, therefore, treated PolyQ35::YFP transgenic animals for 12 hr with the four different translation inhibitors, followed by bortezomib. As expected, treating PolyQ::YFP animals with bortezomib caused extensive protein aggregation. As before, the inhibition of TI exacerbated bortezomib toxicity and was not further quantified. As with *hsf-1(sy441)*, bortezomib treatment also induced the *sma* phenotype in the PolyQ35::YFP transgenic animals, which was almost entirely rescued by anisomycin (*Figure 4E and F*).

We then directly tested the effect of each inhibitor on proteasome activity in both N2 and *hsf-1(sy441)* animals. TE inhibitors did not seem to affect proteasome activity in either strain. In contrast, TI inhibitors reduced N2 proteasome activity compared to DMSO-treated controls, providing a rationale for their increased toxicity in our proteasome survival assay (*Figure 4G*). *hsf-1(sy441)* mutants showed significantly decreased proteasome activity compared to N2 controls, which was not further reduced by TI inhibitors in *hsf-1(sy441)*.

Therefore, inhibition of TI exacerbates bortezomib-induced proteasome stress, while TE inhibition provides limited protection. However, TE inhibitors were highly protective in the context of reduced folding capacity. We concluded that there are distinct protective mechanisms that are employed depending on the way translation is inhibited and the type of stress challenging the animal.

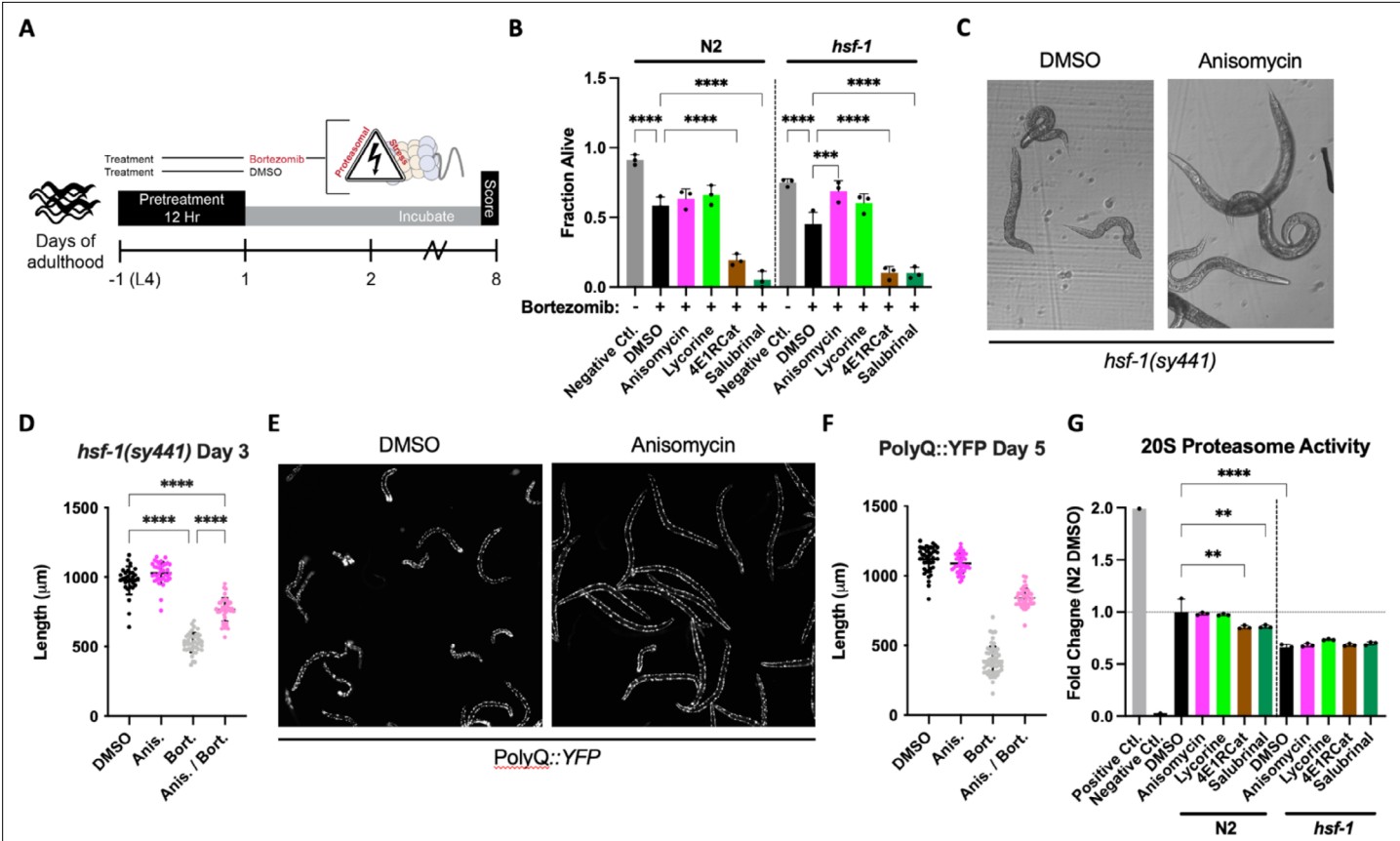

**Figure 4.** Elongation inhibitors protect *C. elegans* from proteasomal stress independent of *hsf-1*. (**A**) Worms were pre-treated for 12 hr with DMSO or indicated inhibitors, followed by bortezomib (75 µM) treatment. The animals were then incubated with the combined treatment for 8 days and scored as alive/dead based on movement. (**B**) TE inhibitor treatment improved morphological features (not shown) and provided limited protection from bortezomib-induced proteotoxicity in N2 animals. TI inhibitor treatment enhanced toxicity. In *hsf-1(sy441)* animals, TE inhibitors protected from bortezomib-induced proteotoxicity, while TI inhibitors continued to sensitize worms to proteotoxicity. Data are displayed as mean ± SD and ****=p < 0.0001 by one-way ANOVA with Dunnet's multiple comparisons test. Total of three independent experiments. (**C**) Representative brightfield images of day 3 *hsf-1(sy441)* animals show anisomycin pre-treatment prevented the *sma* phenotype observed to be caused by proteasomal inhibition. (**D**) Measured length of *hsf-1(sy441)* worms at day 3 of adulthood. Anisomycin treatment almost completely rescued the *sma* phenotype induced by bortezomib. Data are displayed as mean ± SD and ****=p < 0.0001 by a one-way ANOVA with Šídák multiple comparisons test. 30–42 animals per condition. Total of three independent experiments. (**E**) Representative fluorescent images of PolyQ worms treated with anisomycin at day 5. Bortezomib treatment caused morphological defects in animals (left panel), and anisomycin pre-treatment prevented these pathological defects (right panel). (**F**) Measured length of PolyQ worms at day 5 of adulthood. Anisomycin treatment almost entirely rescued the small (*sma*) phenotype induced by bortezomib. Data are displayed as mean ± SD and ****=p < 0.0001 by one-way ANOVA with Dunnet's multiple comparisons test. 42–57 animals per condition. Total of three independent experiments. (**G**) TI inhibitor treatment significantly reduced 20S proteasomal activity in N2 lysate. Compared to the wild-type, the proteasomal activity was lower in *hsf-1(sy441)* mutants, but TI inhibitors did not further reduce it. Positive control: 5 µL of 20S proteasome positive control (Chemicon Part No. 90205). Negative control: N2 lysate treated with 25 µM lactacystin, a 20S proteasome inhibitor (Chemicon Part No. 90208). Data are displayed as mean ± SD where **=p < 0.01 and ****=p < 0.0001 by one-way ANOVA with Šídák multiple comparisons test. Three biological replicates.

The online version of this article includes the following source data and figure supplement(s) for figure 4:

**Source data 1.** Quantification of survival of N2 and *hsf-1(sy441)* treated with bortezomib.

**Source data 2.** Uncropped brightfield micrographs.

**Source data 3.** Quantification of *hsf-1(sy441)* length.

**Source data 4.** Uncropped fluorescence micrographs.

**Source data 5.** Quantification of polyglutamine (PolyQ) length.

**Source data 6.** Quantification of 20S proteasome assay.

**Figure supplement 1.** Survival of N2 and *hsf-1(sy441)* animals treated with translation inhibitors.

**Figure supplement 1—source data 1.** Quantification of survival of N2 and *hsf-1(sy441)* not treated with bortezomib.

## Lifespan extension by translational elongation and initiation inhibitors is dictated by genetic background

Lowering translation is an established mechanism to extend lifespan and delay aging (*Steffen and Dillin, 2016*; *Anisimova et al., 2018*; *Klaips et al., 2018*; *Hansen et al., 2007*; *Pan et al., 2007*). Furthermore, aging is a well-known driver of protein aggregation. However, to our knowledge, it has never been investigated if the anti-aggregation and anti-aging effects of translational inhibition can be uncoupled and if the mode of translational inhibition influences these phenotypes. Inhibition of translation in wild-type animals using the two TE inhibitors, anisomycin and lycorine, showed no, or only a minor, lifespan extension in N2 animals (*Figure 5A*). In contrast, inhibition of TI by the inhibitors 4E1RCat and salubrinal dose-dependently extended lifespan of N2 animals (*Figure 5B*). This difference was observed despite all four translation inhibitors reducing protein translation to the same extent (*Figure 1B*). Thus, the difference in the effect on lifespan by TE and TI inhibitors cannot be explained by reducing overall protein synthesis alone.

While investigating the effect of inhibiting different modes of translation in our proteasome survival assay, we noticed that anisomycin and lycorine appeared to improve the survival of *hsf-1(sy441)* animals (*Figure 4—figure supplement 1*). We, therefore, tested the ability of all four translation inhibitors to extend lifespan in *hsf-1(sy441)* mutants. As expected, both 4E1RCat and salubrinal failed to extend lifespan in *hsf-1(sy441)* mutants significantly. This result is consistent with genetic work by both the Rodgers and Tavernarakis labs, which showed inhibition of TI extends lifespan dependent on HSF-1 (through *ifg-1* and *ife-2*, respectively) (*Howard et al., 2016*; *Rieckher et al., 2018*).

As *Figure 4—figure supplement 1* hinted, treatment with the TE inhibitors anisomycin and lycorine significantly extended the lifespan of *hsf-1(sy441)* mutants by ~20% (*Figure 5B*). Similar results were obtained for the TE inhibitor cycloheximide (*Figure 5—figure supplement 1*). We interpret this lifespan extension as a partial rescue of the protein folding defect in *hsf-1(sy441)* mutants, as the increase in lifespan did not reach the lifespan of wild-type animals. Taken together, however, our data demonstrate that longevity induced by translation inhibition, whether through targeting TE or TI, subsumes several different mechanisms that lead to longevity.

We next tested the ability of all four inhibitors to reduce age-associated protein aggregation. If inhibition of translation alone is sufficient to reduce protein aggregation independently of any downstream mechanisms, all four inhibitors should reduce age-associated protein aggregation. Conversely, if the reduction of age-associated protein aggregation is closely linked to longevity, then only the TI inhibitors should reduce protein aggregation. We treated day 1, N2 animals with 100 μM of each of the four inhibitors and allowed the animals to age for 8 days, after which we separated proteins based on solubility (*Figure 5C*). We found that only the two TI inhibitors that extended lifespan caused significant decreases in the amount of SDS-insoluble aggregates and that the TE inhibitors failed to do so (*Figure 5D and E*).

Taken together, our data suggest that inhibition of TE rescues proteostasis-compromised animals by the *reduced folding load model* without generating additional protein folding capacity. Separately, inhibition of TI protects and improves longevity by the *selective translation model* that depends on HSF-1.

## Inhibition of translation initiation increases protein concentration in *hsf-1(sy441)* mutants

The canonical model that explains the requirement of HSF-1 to induce longevity by inhibiting TI suggests that blocking TI activates the HSR via HSF-1 that then initiates the transcription of HSP chaperones to increase folding capacity (*Rogers et al., 2011*; *Howard et al., 2016*). In contrast, work in cancer cells showed inhibition of translation to block HSF-1 binding to DNA and to specifically reduce the expression of HSP70 and DNAJA chaperones (*Santagata et al., 2013*), making it difficult to envision how HSF-1 could act as a downstream effector of TI.

We, therefore, tested if HSF-1 could act upstream of TI either by controlling the expression of transcripts that become translated upon inhibition of translation or that HSF-1 (or one of its targets) directly contributes to the proper function of the TI machinery. We treated N2 or *hsf-1(sy441)* animals with the four inhibitors and measured the resulting concentration of newly synthesized proteins. As before, all four inhibitors reduced the concentration of newly synthesized proteins in N2. In contrast, TI inhibitors increased the concentration of newly synthesized proteins by 1.5- to 2-fold in the *hsf-1(sy441)* animals

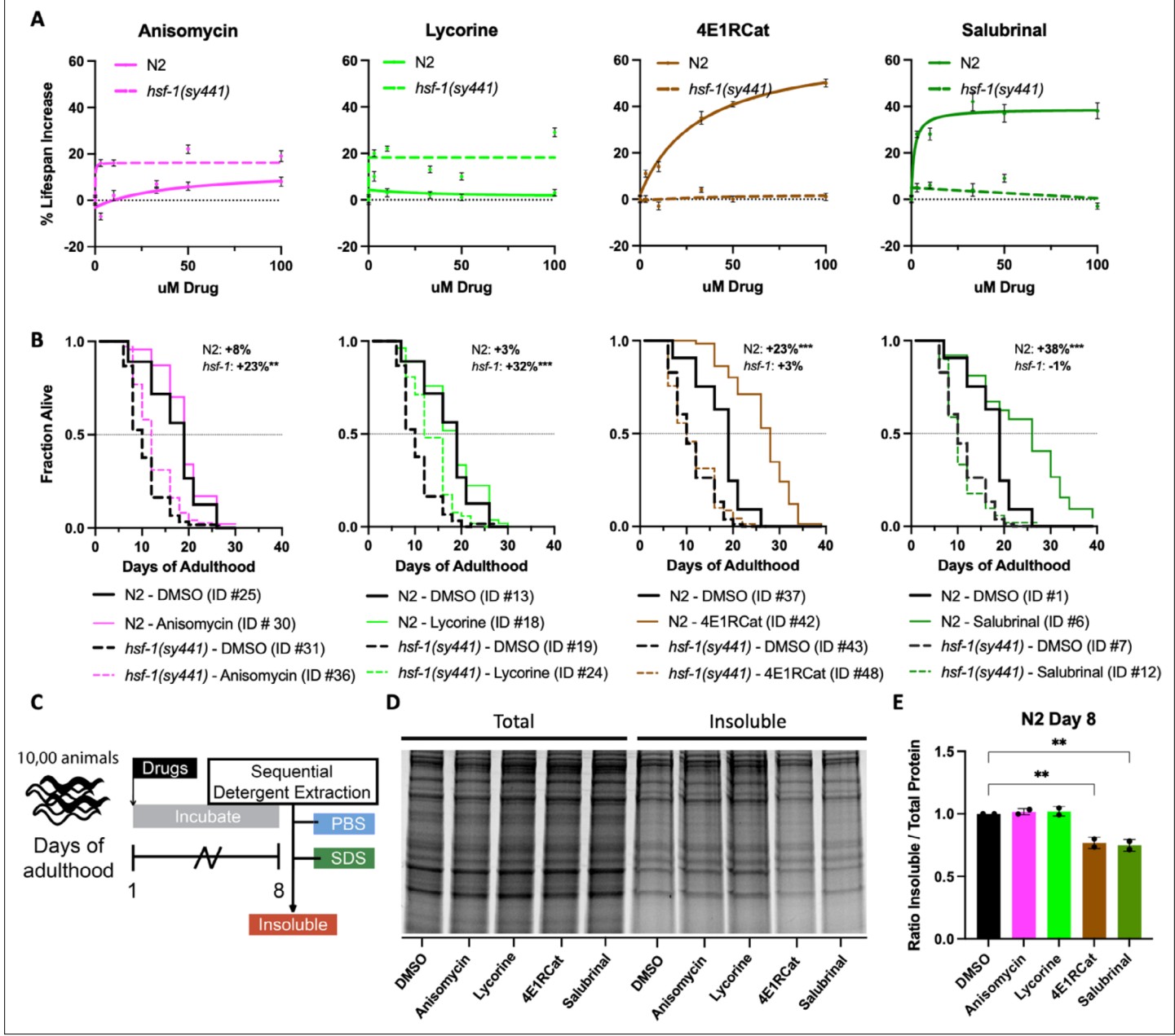

**Figure 5.** Reciprocal lifespan extension by translation inhibitors in N2 and *hsf-1(sy441)* animals. (**A**) Graphs show mean lifespan as a function of translation inhibitor concentration. TI inhibitors increase the lifespan of N2 but not *hsf-1(sy441)* animals, with a maximum effect at 100 μM. TE inhibitors increase the lifespan of *hsf-1(sy441)* but not N2 animals. Error bars indicate ± SEM. See **Supplementary file 1** for the number of animals and repeats. (**B**) Survival curves from representative experiments show the fraction of wild-type (N2, solid line) or *hsf-1* mutant (dashed line) animals when treated with 100 μM of the indicated compound. Black lines indicate DMSO treatment, and colored lines indicate inhibitor treatment. Data are displayed as a Kaplan-Meier survival curve, and significance was determined by the log-rank test. ID # refers to the unique entry within **Supplementary file 1**. (**C**) Experimental strategy for treating animals and isolating detergent-insoluble fractions. 10,000 animals were treated and allowed to age for 8 days before being washed with M9, frozen in liquid nitrogen, and mechanically lysed. Then proteins were extracted from the total lysate based on solubility, and an aliquot from each fraction was run on an SDS-PAGE gel. (**D**) Representative SDS-PAGE gel stained with Sypro Ruby. 4E1RCat and salubrinal reduce insoluble protein at day 8. (**E**) Quantification of two separate experiments shows 4E1RCat and salubrinal significantly reduce insoluble protein in wild-type (**N2**) animals. Data are displayed as mean ± SEM and **=$p < 0.01$ by two-tailed Student's t-test.

The online version of this article includes the following source data and figure supplement(s) for figure 5:

**Source data 1.** Summary of lifespan data used to construct dose-response graphs.

**Source data 2.** Lifespan data used to construct graphs.

*Figure 5 continued on next page*

*Figure 5 continued*

**Source data 3.** Unedited gels.

**Source data 4.** Quantification of insoluble extractions.

**Figure supplement 1.** Cycloheximide, an elongation inhibitor, extends lifespan in *hsf-1(sy441)* but not N2.

**Figure supplement 1—source data 1.** Lifespan data used to construct graphs.

(*Figure 6A and B*). We confirmed that this effect is not due to increased intake of puromycin, as inhibitor treatment did not affect food intake (*Figure 6C*). Furthermore, none of the translation inhibitors induced the HSF-1-activated heat shock reporter *hsp-16.2::GFP* (*Figure 6D*).

To ensure that these results are indeed caused by the inhibition of TI and not an off-target effect, we knocked down the target of 4E1RCat, eIF4G/*ifg-1,* by RNAi in N2 and *hsf-1(sy441)* animals and measured the concentration of newly synthesized proteins. As seen for 4E1RCat, the knockdown of eIF4G/*ifg-1* reduced the protein concentration in N2 but dramatically increased it in *hsf-1(sy441)* mutants (*Figure 6E*). Unfortunately, we could not determine the exact target for salubrinal in *C. elegans*.

To further confirm that the lifespan extension of 4E1RCat is caused by on-target action, we measured lifespan extension in N2 and *ifg-1(cxTi9279)* mutants which carry a splicing defect that results in an overall reduction of eIF4G/*ifg-1* protein (*Morrison et al., 2014*). As we previously observed (*Figure 5A and B*) 4E1RCat treatment of N2 animals significantly extended lifespan (*Figure 6F*). However, no lifespan extension was observed in *ifg-1(cxTi9279)* animals (*Figure 6G*), showing that 4E1RCat extends lifespan by on-target action. Therefore, we concluded that inhibition of eIF4G/*ifg-1* requires HSF-1 to lower the concentration of newly synthesized proteins. Consequently, the beneficial effects of 4E1RCat treatment are not observed in *hsf-1(sy441)* mutants as inhibition of eIF4G/*ifg-1* fails to lower protein concentrations in this strain.

## Discussion

A substantial body of genetic and biochemical work suggests that lowering the concentration of newly synthesized proteins reduces the load on the proteostasis machinery. We asked if these effects could be pharmacologically replicated to exploit them for therapeutic purposes in disease models (*Solis et al., 2018*; *Schubert et al., 2018*).

We set out to study how pharmacological inhibition of TI or TE improves proteostasis and increases longevity in adult *C. elegans*. We intended to compare the *selective translation* model, in which the selective reduction of specific proteins and the selective translation of others protect from proteotoxicity to the *reduced folding load model*, in which the non-specific reduction of the concentration of newly synthesized proteins is protective.

We compared the non-selective lowering of protein concentration by TE inhibitors to the selective lowering of initiation-dependent proteins by TI inhibitors. Our studies reveal that pre-existing proteotoxicity, as might be observed in a disease state, dictates the mode of translation inhibition necessary to restore proteostasis. As a result, the mode of translation inhibition, either inhibiting TI or TE, results in distinct and sometimes opposing outcomes for the health of the animals.

The most striking dichotomy was seen when pre-existing proteotoxicity—caused by the lack of either HSF-1 or the expression of the aggregation-prone PolyQ protein—was combined with the inhibition of the proteasome. These combined insults led to increased protein aggregation, shrinking body size, and death of the animals (*Figure 4*). Inhibition of TE almost entirely rescued these phenotypes. In sharp contrast, inhibition of TI exacerbated proteotoxicity and increased mortality. The effects were consistent for both TE and TI inhibitors despite structural differences and different targets of the TI inhibitors.

The second example of this dichotomy was the reciprocal ability of TI and TE inhibitors to extend the lifespan of N2 wild-type animals or *hsf-1(sy441)* mutant animals, with each mode of inhibition only able to extend the lifespan in one of the strains. Overall, TE inhibition effectively rescued proteostasis in animals with pre-existing proteotoxicities, such as a compromised HSR, reduced proteasome activity, or the expression of aggregation-prone proteins. Conversely, inhibition of TI did little to rescue pre-existing proteotoxicity or worsen it but proved effective at preventing emerging proteotoxicity,

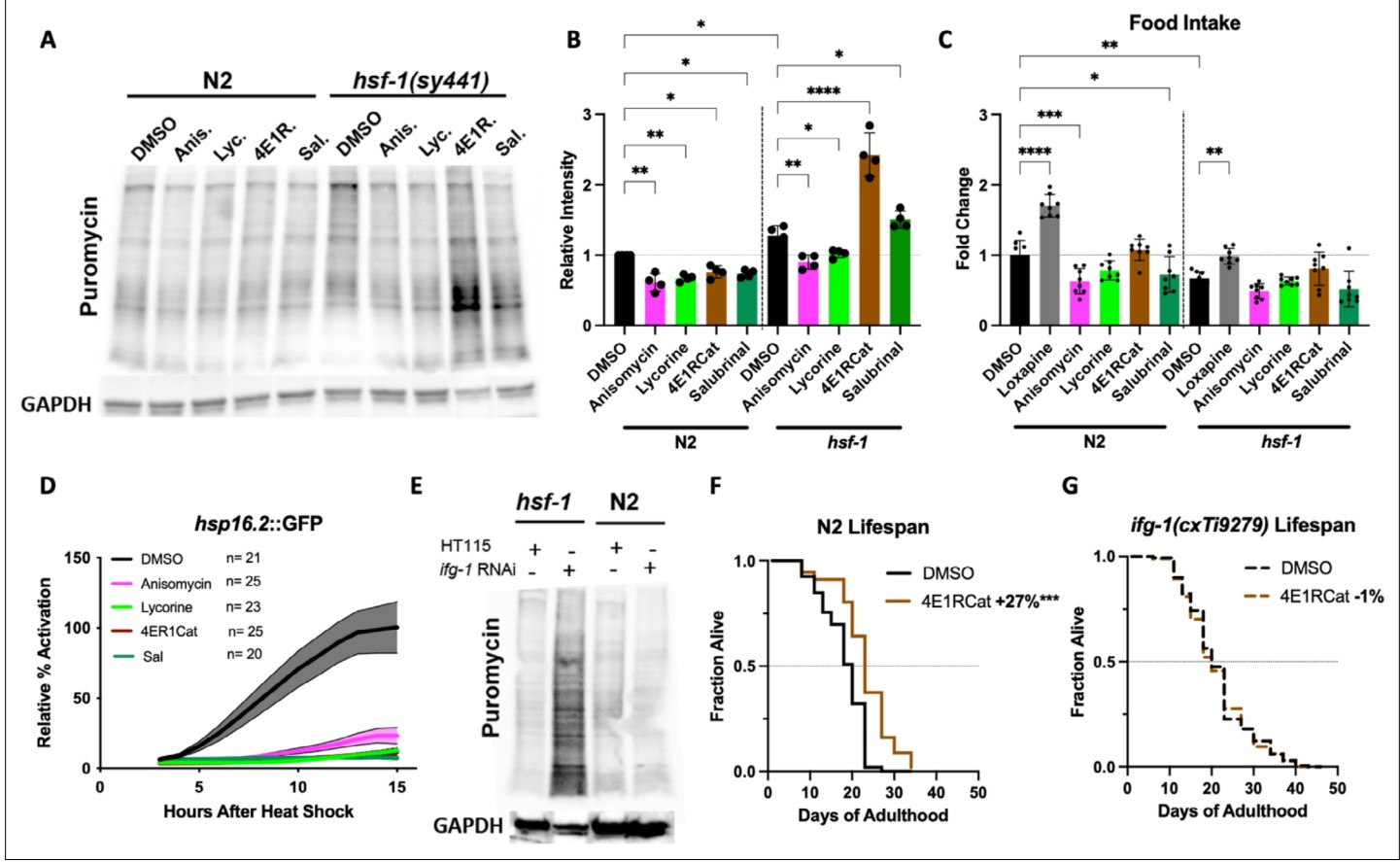

**Figure 6.** Inhibition of eIF4G/IFG-1 requires HSF-1 to lower the concentration of newly synthesized proteins. (**A**) *C. elegans* were treated with solvent (DMSO) or the indicated inhibitors (100 µM) for 12 hr, followed by a 4 hr puromycin incorporation in both N2 and *hsf-1(sy441)* background, which was immunoblotted against using an anti-puromycin antibody. *hsf-1(sy441)* mutant animals exhibited increased puromycin incorporation compared to N2 animals, which was further increased by both TI inhibitors 4E1RCat and salubrinal. (**B**) Quantification of four independent SUrface SEnsing of Translation (SUnSET) experiments in (A). Significance was determined by one-way ANOVA with Šídák multiple comparisons test where *=p ≤ 0.05, **=p ≤ 0.01, and **** p≤0.0001. Error bars indicate mean ± SD from four independent trials. (**C**) Food intake was quantified relative to DMSO-treated N2 controls measuring bacterial clearance from day 1 to day 4 of adulthood. Anisomycin and salubrinal significantly decrease food intake in wild-type N2. In *hsf-1(sy441)*, no inhibitor statistically changes food intake. Loxapine was used as a positive control in both genotypes. Significance was determined by one-way ANOVA with Šídák multiple comparisons test where *=p ≤ 0.05, **=p ≤ 0.01, and ****p≤0.0001. Error bars indicate mean ± SD from four independent trials. Total of eight independent experiments. (**D**) Translation inhibitors suppressed the heat shock response (HSR) as measured by *hsp-16.2*::GFP fluorescence assay. After incubation with the inhibitors for 4 days, followed by a 1 hr heat shock (HS) at 36°C, little to no increase in GFP expression was observed for each inhibitor, indicating that all inhibitors block HSR activation at the tested concentration (100 µM). Lines indicate mean, and shading indicates 95% CI. Representative of three independent experiments. (**E**) N2 and *hsf-1(sy441) C. elegans* were fed HT115 empty vector or RNAi against *ifg-1. ifg-1* depletion increased puromycin incorporation in *hsf-1* mutants but decreased incorporation in N2 animals, similar to 4E1RCat treatment. GAPDH was used as a loading control. Immunoblot is representative of three independent experiments. (**F**) 4E1RCat significantly increases lifespan in N2 animals. Data are displayed as a Kaplan-Meier survival curve, and significance is determined by the log-rank test. See ID #111 and #112 for details in ***Supplementary file 1***. (**G**) 4E1RCat does not increase lifespan in *ifg-1(cxTi9279)* animals. See ID #93 and #94 for details in ***Supplementary file 1***. Data are displayed as a Kaplan-Meier survival curve, and significance is determined by the log-rank test.

The online version of this article includes the following source data for figure 6:

**Source data 1.** Unedited western blots.

**Source data 2.** Quantification of western blots.

**Source data 3.** Quantification of food intake.

**Source data 4.** Quantification of *hsp-16.2::GFP* reporter activation.

**Source data 5.** Unedited western blots with Coomassie staining of the membrane.

**Source data 6.** Lifespan data used to construct graphs.

**Source data 7.** Lifespan data used to construct graphs.

for example, with age. Notably, both anisomycin and cycloheximide have previously been reported to extend lifespan in wild type animals (*Tarkhov et al., 2019*; *Takauji et al., 2016*) but these studies were conducted in either non-fully reproductive adults or the exact conditions were not specified (i.e. temperature, whether OP50 feeding bacteria was alive/dead and developmental stage; *Takauji et al., 2016*). In contrast, we focused on the effects of inhibitor treatment on the lifespan of fully reproductive adult animals—thereby eliminating any partial dependency on developmental effects. Overall, our findings establish that inhibition of TI or TE has distinct consequences for proteostasis and that their mechanisms to control the concentration of newly synthesized proteins can be genetically separated.

We considered the following reasons to explain the different consequences for proteostasis. The increased mortality observed after combining bortezomib with TI inhibitors is likely to be the result of the lower expression of HSP90/*daf-21* and HSP70/*hsp-1*, whose translation depends on eIF4G/*ifg-1* (*Rogers et al., 2011*) and the ability of both treatments to lower proteasome activity independently (*Figure 4*). The lower proteasome activity induced by the TI inhibitors is likely to enhance the proteotoxicity of bortezomib. The lowered proteasome activity observed upon inhibition of TI was surprising as the short-term inhibition of translation increases proteasome activity (*Zhao et al., 2015*). However, compared to most other studies, we chose to reduce translation by only ~40%, as such a reduction is more likely to be tolerated chronically if translation inhibitors could be developed into therapeutics. In contrast, TE inhibitor treatment alone did not lower proteasome activity. More importantly, lowering the concentration of newly synthesized proteins by inhibiting TE rescued most bortezomib-induced proteotoxic effects. Thus, long-term inhibition of TI or TE differentially affects proteasomal activity and its proteotoxic effects.

The most unequivocal difference was the dependency of TI inhibitors on HSF-1, which was not observed for TE inhibitors. This dependency of the TI inhibitors on HSF-1 was initially explained by a model in which inhibition of TI activates the HSR via HSF-1. This model has been previously proposed for eIF4G/*ifg-1* (*Rogers et al., 2011*; *Howard et al., 2016*).

This initial model is still likely valid, but our study shows that HSF-1 has a more immediate role. HSF-1, or one of its targets, is necessary for the TI machinery to control protein synthesis. *Selective translation* can only occur if TI itself is inhibited. As seen in *Figure 6*, inhibition of TI in the absence of HSF-1 increases, rather than decreases, the amount of newly synthesized proteins. This surprising increase in newly synthesized proteins upon 4ER1Cat treatment was not an off-target effect as it was replicated by RNAi-mediated knockdown of eIF4G/*ifg-1* in an HSF-1 mutant background. Hence, HSF-1 activity directly or indirectly modulates the ability of TI components to control the concentration of newly synthesized proteins.

At the outset, we intended to compare the *selective translation* model to the *reduced folding load model* by comparing their effects on proteostasis under conditions in which inhibition of translation resulted in the same concentration of newly synthesized proteins, irrespective of the mode of translation inhibition. We observed apparent differences in proteostasis effects, supporting the *reduced folding load model* for inhibiting TE and the *selective translation model* for inhibiting TI.

The inability of TI inhibitors to reduce the concentration of newly synthesized proteins in *hsf-1(sy441)* mutants and the inability to extend their lifespan shows that lowering the concentration of newly synthesized proteins is necessary for the beneficial effects. On the other hand, the finding that TE inhibitors protect from proteotoxic stress but do not extend lifespan shows that lowering the concentration of newly synthesized proteins is sufficient to protect from proteotoxic stress but is not sufficient to extend lifespan in wild-type, which appears to require *selective translation*.

## Ideas and speculation

Our data show that HSF-1 or one of its targets controls the ability of TI to change the concentration of newly synthesized proteins. Previous work noted a surprising requirement for HSF-1 in the protection from ER stress elicited by the RNAi-mediated knockdown of eIF4G/*ifg-1* (*Howard et al., 2016*). The authors note that further investigations will be required to determine the direct link between enhanced HSR amelioration of proteotoxicity and the endoplasmic reticulum. Our results provide a compelling explanation for their findings, in that the lack of HSF-1 abolishes the ability of *ifg-1* RNAi to lower the concentration of newly synthesized proteins and thus the protection from tunicamycin toxicity.

In the context of TI inhibition, HSF-1 (or one of its targets) may influence the concentration of newly synthesized proteins by either modulating protein degradation or synthesis. Our data show that proteasome activity and, by extension, degradation are lowered by both inhibition of TI and lack of HSF-1. Hence, combining both may indirectly increase the concentration of newly synthesized proteins through reduced degradation. However, inhibition of TI did not further decrease the already low proteasome activity in HSF-1 mutants (*Figure 4*). Furthermore, salubrinal and 4ER1Cat lowered proteasome activity by the same amount, yet 4ER1Cat treatment increased the concentration of newly synthesized proteins much more in the HSF-1 background (*Figure 6B and E*). Thus, we concluded a model in which HSF-1 promotes degradation in response to the inhibition of the TI machinery less likely.

Alternatively, HSF-1 (or one of its targets) cooperates with TI factors to form a checkpoint that prevents random unregulated translation. The TI factors are part of the checkpoint but also initiate regulated translation by giving ribosomes access to mRNA. Thus, when TI factors are inhibited or knocked down, HSF-1 or one of its targets maintains the checkpoint preventing uncontrolled translation. Without any TI activity, maintaining the checkpoint reduces protein synthesis. In the *hsf-1(sy441)* mutant, the TI machinery still prevents uncontrolled translation but less effectively, leading to a slight but detectable increase in newly synthesized proteins in HSF-1 mutants (*Figure 6A*). Once the TI and HSF-1 are removed, ribosomes gain access to many mRNAs leading to unregulated translation and a dramatic increase in newly synthesized proteins.

Both the degradation and the cooperative checkpoint model explain the observed results but need to invoke a speculative connection between HSF-1 to either TI or protein degradation. Our results unexpectedly couple the ability of the TI machinery to control the concentration of newly synthesized proteins to HSF-1, thereby explaining its requirement. However, our results do not provide a mechanism by which HSF-1 couples the concentration of newly synthesized proteins to TI.

## Materials and methods

### Key resources table

| Reagent type (species) or resource | Designation | Source or reference | Identifiers | Additional information |
|---|---|---|---|---|
| Strain, strain background (*Caenorhabditis elegans*) | N2 | *Caenorhabditis* Genetics Center (CGC) | RRID:WB-STRAIN:WBStrain00000003 | Wild-type (Bristol) |
| Strain, strain background (*Caenorhabditis elegans*) | CL2070 | CGC | RRID:WB-STRAIN:WBStrain00005096 | *dvIs70 [hsp-16.2p::GFP+rol-6(su1006)]* |
| Strain, strain background (*Caenorhabditis elegans*) | AM140 | CGC | RRID:WB-STRAIN:WBStrain00000182 | *rmIs132Punc-54::q35::yfp* |
| Strain, strain background (*Caenorhabditis elegans*) | KX54 | CGC | RRID:WB-STRAIN:WBStrain00024080 | *ifg-1(cxTi9279)* |
| Strain, strain background (*Caenorhabditis elegans*) | PS3551 | CGC | RRID:WB-STRAIN:WBStrain00007673 | *hsf-1(sy441)* |
| Genetic reagent (*Caenorhabditis elegans*) | *myo3p*::GFP-IRES-tdTomato | This paper | | Adapted from: DOI: 10.2144/000113821 See Materials and methods, Method for making bi-cistronic vector |
| Antibody | Anti-puromycin (Mouse monoclonal) | MilliporeSigma | Cat#: MABE343 RRID:AB_2566826 | 1:5000 |
| Antibody | Anti-GAPDH (Rabbit polyclonal) | Proteintech | Cat#: 1094-1-AP RRID:AB_2263076 | 1:5000 |
| Antibody | Anti-myc (Mouse Monoclonal) | Cell Signaling | Cat#: 2276S RRID:AB_331783 | 1:2000 |
| Antibody | Anti-mouse—HRP (secondary) | Cell Signaling | Cat#: 7076S, RRID:AB_330924 | 1:5000 |
| Antibody | Anti-rabbit—HRP (secondary) | Cell Signaling | Cat#: 7074S, RRID:AB_2099233 | 1:5000 |
| Commercial Assay or kit | 20S Proteasome Activity Assay Kit | Sigma-Aldrich | Cat#: APT280 | |
| Commercial Assay or kit | SYPRO Ruby Protein Gel Stain 1× | Bio-Rad | Cat#: 1703125 | |
| Chemical compound, drug | Anisomycin | MedChemExpress | Cat#: HY-18982 | |

*Continued on next page*

*Continued*

| Reagent type (species) or resource | Designation | Source or reference | Identifiers | Additional information |
|---|---|---|---|---|
| Chemical compound, drug | Lycorine hydrochloride | Combi-Blocks | Cat#: QW-2476 | |
| Chemical compound, drug | 4E1RCat | MedChemExpress | Cat#: HY-14427 | |
| Chemical compound, drug | Puromycin | Sigma-Aldrich | Cat#: P8833 | |
| Chemical compound, drug | Salubrinal | MedChemExpress | Cat#: HY-15486 | |
| Chemical compound, drug | Bortezomib, free base | LC Laboratories | Cat#: B-1408 | |

## Lead contact and materials availability

Michael Petrascheck is the Lead Contact and may be contacted at pscheck@scripps.edu. This study did not generate new unique reagents; however, the natural product Amicoumacin C was obtained as a gift from Dr Shigefumi Kuwahara, Ph.D. (Tohoku University). This reagent is not available without total chemical synthesis.

## *C. elegans* strains

The Bristol strain (N2) was used as the wild-type strain. In addition, the following worm strains used in this study were obtained from the Caenorhabditis Genetics Center (CGC; Minneapolis, MN, USA): CL2070 [dvIs70 [*hsp*-16.2p::GFP+*rol-6(su1006)*]], AM140 [rmIs132[P*unc-54*::q35::yfp]], KX54 [*ifg-1(cxTi9279)*] and PS3551 [*hsf-1(sy441)*].

## Method details

### Worm maintenance

1000–2000 age-synchronized animals were plated into 6 cm culture plates with liquid medium (S-complete medium with 50 mg/mL carbenicillin and 0.1 mg/mL fungizone [amphotericin B]) containing 6 mg/mL X-ray irradiated *Escherichia coli* OP50 ($1.5×10^8$ colony-forming units [cfu]/mL, carbenicillin resistant to exclude growth of other bacteria), freshly prepared 4 days in advance, as previously described (*Solis and Petrascheck, 2011*), and were maintained at 20°C. The final volume in each plate was 7 mL. To prevent self-fertilization, FUDR (5-fluoro-2′-deoxyuridine, 0.12 mM final) (Sigma-Aldrich, Cat#: 856657) was added 42—45 hr after seeding. At the late L4 stage, either DMSO/drug treatment (100 µM unless otherwise stated) was added to each strain.

### RNAi

RNAi of *ifg-1* was preformed using the feeding vector L4440 to express dsRNA of the respective genes, received as a gift from the Hansen lab. HT115 bacteria were as OP50 bacteria above, with the exception that prior to harvesting, 2 mM IPTG (isopropyl β-D-1-thiogalactopyranoside) was added, and the suspension was allowed to grow for 4 additional hours to induce dsRNA expression. Animals were grown and synchronized in liquid culture as previously described, being fed the relevant RNAi bacteria.

### SUnSET to analyze the effectiveness of translation inhibitors in *C. elegans*

Day 1 adult N2 worms were bleached, and eggs were allowed to hatch in S-complete by shaking overnight. On the next day, 12,000 L1 worms were seeded in a 15 cm plate containing a total volume of 30 mL S-complete with 6 mg/mL OP50 bacteria, 50 µg/mL carbenicillin, and 0.1 µg/mL amphotericin B. Six mL of 0.6 mM FUDR were added to worms at L4 stage in each plate. 100 µM translation inhibitor was added to worms 2 hr after adding FUDR. After 12 hr, worms were transferred into a 15 mL corning tube containing a total volume of 5 mL S-complete with 750 µL 6 mg/mL OP50 bacteria, 0.5 mg/mL puromycin, and 100 µM translation inhibitors. After rotating the corning tubes for 4 hr, worms were collected into 2 mL cryotubes by washing them with M9 once and with cold PBS three times. Worms were flash-frozen in liquid nitrogen and subsequently broken with a beak mill homogenizer (Fisherbrand). Protein concentrations were determined by the Bradford protein assay. 50 mg protein from each sample was loaded for western blot analysis using antibodies against puromycin

(Millipore, MABE343) and GAPDH (Proteintech, 10494-1-AP). Antibodies were diluted 1:5000 in 5% non-fat milk in TBST.

## Method for making bi-cistronic vector

The body wall muscle-specific promotor of the bi-cistronic vector is from 2 kb upstream of the start codon of gene myo-3. GFP coding sequence is after the promotor. The IRES element is from 285 bp upstream of the hsp-3 start codon (*Li and Wang, 2012*). The Tdtomato coding sequence is after the IRES fragment. A 3Xmyc fragment is inserted in front of the tdtomato sequence. All fragments were ligated together by Gibson Assembly method (New England Biolabs).

## Determination of GFP and tdTomato fluorescence

Day 1 adult bi-cistronic reporter worms were bleached, and eggs were allowed to hatch in S-complete by shaking overnight. On the next day, 12,000 L1 worms were seeded in a 15 cm plate containing a total volume of 30 mL S-complete with 6 mg/mL OP50 bacteria, 50 µg/mL carbenicillin, and 0.1 µg/mL amphotericin B. Six mL of 0.6mM FUDR was added, and two hours later, the animals were treated with the indicated translation inhibitor at a final concentration of 100 µM or DMSO only. At the L4 stage, 500 GFP-positive animals per condition were sorted using the COPAS Biosorter (Union Biometric). The integral values for fluorescence on a corresponding channel (i.e. green or red) were used to determine simultaneous red and green measurements for each animal using the standard BioSorter software. Integral values are determined automatically by the BioSorter by integrating each signal when the threshold signal is above the threshold value. In essence, it determines intensity over the length of the worm to account for variations in size.

## Thermotolerance

Age-synchronized N2 or PS3551 [*hsf-1(sy441)*]] animals were prepared as above in 6 cm culture plates and treated with water or 100 µM lycorine/anisomycin on day 1. On day 4, 25–35 animals were transferred to 6 cm NGM plates in triplicate for each condition and were transferred to the non-permissive temperature of 36°C. Every hour, survival was scored by lightly touching animals with a worm pick and scoring for movement.

## *C. elegans* insoluble protein extraction

10,000 N2 worms were sorted into a 15 cm liquid culture dish using the COPAS Biosorter (Union Biometrica). For heat shock-induced aggregation experiments, worms were treated with either DMSO or anisomycin (100 µM) for 12 hr on day 1 of adulthood, then subjected to a 2 hr heat shock at 36°C. After 12 hr of recovery, the animals were washed three times with S-complete buffer, once with PBS, and then flash-frozen in liquid nitrogen. 500 µL of cold lysis buffer (20 mM Tris base, 100 mM NaCl, 1 mM $MgCl_2$, pH = 7.4, with protease inhibitors [Roche, 11836153001]) was added, and animals homogenized mechanically. An aliquot of this total lysate was saved. In an ultracentrifuge tube, two volumes of SDS Extraction buffer (20 mM Tris base, 100 mM NaCl, 1 mM $MgCl_2$, pH = 7.4, with protease inhibitors, and 1% SDS) were added to 1 volume of total lysate and was centrifuged at 20,000 × *g* for 30 min. The extraction was repeated two times to remove all SDS-soluble proteins. The remaining insoluble pellet was suspended briefly in 20 µL urea buffer (8 M urea, 50 mM DTT, 2% SDS, 20 mM Tris base, pH = 7.4) and sonicated. 18 µL of the insoluble suspension was added to 6 µL 4× Laemmli buffer (Bio-Rad, #161-0747) supplemented with 10% 2-mercaptoethanol (Sigma, 60-24-2) and boiled for 5 min, then directly loaded onto SDS-PAGE gel (Bio-Rad, 4569033). Gels were stained with Sypro Ruby according to the manufacturer's directions. Gels were quantified in ImageJ by dividing the integrated intensity of each full insoluble lane by the integrated intensity of the corresponding full total protein lane after subtracting a similar area background lane, then normalizing to the DMSO control.

For age-associated protein aggregation experiments, the above was repeated with the following changes: Day 1 worms were treated with 100 µM of each compound and allowed to age in liquid culture until day 8 of adulthood. Following lysis, worms were washed several times with PBS to remove soluble protein.

## Proteasome dysfunction assay—survival

Animals were prepared as above in 96-well plates. At the late L4 stage, animals were pre-treated with DMSO or 100 µM anisomycin. After 12 hr, the animals were treated with 75 µM bortezomib. On day 8 of adulthood, the percentage of animals alive was determined by movement in liquid culture.

## Proteasome dysfunction assay—worm length

Animals were prepared as above in 6 cm liquid culture dishes. At the late L4 stage, animals were pre-treated with DMSO or 100 µM anisomycin. After 12 hr, the animals were treated with 75 µM bortezomib. Body length was measured on day 3 for *hsf-1(sy441)* or day 5 for PolyQ using the 10× objective with the ImageXpress Micro XL and Metaexpress microscopy software.

## 20S proteasome activity assay

The Chemicon 20S Proteasome Activity Assay Kit (Cat. No APT280) was used according to the manufacturer's instructions. In short, 10,000 age-synchronized animals were grown in liquid culture on 10 cm plates. DMSO or inhibitors were added on day 1. On day 5, animals were collected and washed 3× with cold DPBS, then once with 1× assay buffer. Worms were flash-frozen in liquid nitrogen. After three biological replicates were harvested in this way, the frozen animals were broken open with a beak mill homogenizer (Fisherbrand). Protein concentrations were determined by the Bradford protein assay. 200 µg of sample was loaded with assay mixture into a 96-well plate and incubated for 1 hr at 37°C. Fluorescence was measured on the Tecan Safire II with a 380/460 nm filter set. Positive control: 5 µL of 20S proteasome positive control (Chemicon Part No. 90205). Negative control: N2 lysate treated with 25 mM lactacystin, a 20S proteasome inhibitor (Chemicon Part No. 90208).

## Heat shock-induced aggregation and stress response

CL2070 [dvIs70 [*hsp-16.2p*::GFP+*rol-6*(su1006)]] or AM140 [rmIs132[Punc-54::q35::yfp]] age-synchronized animals were treated with DMSO or 100 µM drug at late L4. 4–12 hr later, on day 1 of adulthood, 1.5 mL of the treated animals were transferred from liquid culture into an Eppendorf tube, washed twice with S-complete, pelleted, and then transferred to 6 cm NGM plates using S-complete. Once the animals were completely dry on the NGM plate, they were transferred to a 36°C incubator, plates upside down, for 1–2 hr.

## Hydrogel mounting

Animals were washed from NGM plates using 0.2% HHPPA (2-hydroxy-4'-(2-hydroxyethoxy)-2-methylpropiophenone) (CAS[106797-53-9]) dissolved in S-complete into a 2 mL Eppendorf tube. Animals were washed twice with 0.2% HHPPA, then suspended in 0.3 mL 0.2% HHPPA. 2.5 µL of this solution was seeded into a single well of a 384-well plate containing 2.5 µL 30% PEG-DA (polyethylene glycol diacrylate, MW = 4000, Polysciences, Cat#: 15246-1) in S-complete. After 5 min, to allow the solutions to diffuse, the animals were immobilized by subjecting the 384-well plate UV light using a routine laboratory gel viewer (UVP Dual-Intensity Ultraviolet Transilluminator, high intensity) for 30 s. 45 µL S-complete buffer was added on top to prevent desiccation. In general, each well contained 5–10 worms.

## Imaging and analysis

Time-lapse brightfield and fluorescence images were taken with a 10× objective using the ImageXpress Micro XL over 15 hr. The number of PolyQ aggregates, or total YFP fluorescence, in the whole worm, was determined by analyzing images using a custom pipeline created in CellProfiler.

## Lifespan assay

Age-synchronized *C. elegans* were prepared in a liquid medium, as described above, and seeded into flat-bottom, optically clear 96-well plates (Corning, 351172) containing 150 µL total volume per well, as previously described (*Clay and Petrascheck, 2020*). Plates contained ~10 animals per well in 6 mg/mL γ-irradiated OP50. Age-synchronized animals were seeded as L1 larvae and grown at 20°C. Plates were covered with sealers to prevent evaporation. To prevent self-fertilization, FUDR (0.12 mM final)

was added 42—45 hr after seeding. Drugs were added on day 1 of adulthood. DMSO was kept to a final concentration of 0.33% vol/vol when used.

## Quantification and statistical analysis

### Aggregation and induction of the HSR

The number of n represents the total number of animals over three individual experiments. For the paired-time-lapse data generated, we chose to depict the 95% confidence interval, calculated by GraphPad Prism, to show differences in treatment.

### Quantification of western blot

To determine the relative intensities of each blot, the integrated intensity (pixel intensity divided by pixel area) was measured for each full lane using ImageJ. A similar-sized band with no signal was used to subtract the background, and then each intensity normalized to its corresponding GAPDH loading control. Finally, each band's integrated intensity was normalized to the wild-type DMSO control for quantification and statistics. Significance was determined by one-way ANOVA with Šídák multiple comparisons test.

### Percent survival—thermotolerance assay

The number of n represents the total number of animals over the three individual replicates shown as the average percentage survival and SEM, calculated using GraphPad Prism. Significance was determined by using a row-matched two-way ANOVA with Šídák multiple comparisons test.

### Worm length—proteasome dysfunction assay

The number of n represents the total number of animals whose length was measured in one experiment. Depicted are the mean and standard deviation calculated using GraphPad Prism. Significance was determined by the two-tailed unpaired t-test. Similar results were observed across three independent experiments.

### Lifespan assay

Survival was scored manually by visually monitoring worm movement using an inverted microscope thrice weekly. Statistical analysis was performed using the Mantel-Haenzel version of the log-rank test as outlined in *Petrascheck and Miller, 2017*.

## Code availability

The software used in this study (Cell Profiler) is available at https://cellprofiler.org/.

## Acknowledgements

We would like to acknowledge Drs Anabel Perez-Gomez, Sarah Ly, Jin Lee, Caroline Kumsta, and Malene Hansen for input into the manuscript; Alan To for technical assistance; and Dr Shigefumi Kuwahara (Tohoku University) for providing Amicoumacin C Grants supported this work to MP from the NIH (DP2 OD008398, R21NS107951, R01AG067331), and the Glenn Foundation. KC was funded by the Dorris Neuroscience Scholar Fellowship. Some strains were provided by the CGC, funded by the NIH Office of Research Infrastructure Programs (P40 OD010440).

## Additional information

### Competing interests

Khalyd J Clay, Michael Petrascheck: is a scientific founder and advisor to Cyclone Therapeutics, Inc, a biotech company developing therapeutics targeting translation. The other authors declare that no competing interests exist.

## Funding

| Funder | Grant reference number | Author |
| --- | --- | --- |
| National Institutes of Health | R21NS107951 | Michael Petrascheck |
| National Institute on Aging | R01AG067331 | Michael Petrascheck |
| The Glenn Foundation | | Michael Petrascheck |
| Dorris Neuroscience Scholar Fellowship | | Khalyd J Clay |

The funders had no role in study design, data collection and interpretation, or the decision to submit the work for publication.

## Author contributions

Khalyd J Clay, Conceptualization, Data curation, Formal analysis, Funding acquisition, Validation, Investigation, Visualization, Methodology, Writing – original draft, Writing – review and editing; Yongzhi Yang, Formal analysis, Investigation, Methodology, Writing – review and editing; Christina Clark, Investigation; Michael Petrascheck, Conceptualization, Supervision, Funding acquisition, Investigation, Writing – original draft, Project administration, Writing – review and editing

## Author ORCIDs

Khalyd J Clay ⓘ https://orcid.org/0000-0003-1381-5295
Yongzhi Yang ⓘ http://orcid.org/0000-0002-9713-0009
Christina Clark ⓘ http://orcid.org/0000-0002-5389-4373
Michael Petrascheck ⓘ https://orcid.org/0000-0002-1010-145X

## Decision letter and Author response

Decision letter https://doi.org/10.7554/eLife.76465.sa1
Author response https://doi.org/10.7554/eLife.76465.sa2

# Additional files

## Supplementary files

• Supplementary file 1. Summary of lifespan studies. All experiments were conducted with dead, γ-irradiated bacteria (OP50) and conducted in a 96-well liquid culture.

• Transparent reporting form

## Data availability

All data generated or analysed during this study are included in the manuscript and supporting file.

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
