## [Editor Report]

Inhibition of translation has been found as a conserved intervention to extend lifespan across a number of species. In this work, the authors systematically investigate the similarities and differences from pharmacological inhibition of protein synthesis at the initiation or elongation steps on longevity and stress resistance. These experiments are important for conceptualizing how translation inhibition actually extends lifespan and promotes proteostasis.

---

## [Decision Letter]

**Decision letter after peer review:**

Thank you for submitting your article "Proteostasis is differentially modulated by inhibition of translation initiation or elongation" for consideration by *eLife*. Your article has been reviewed by 3 peer reviewers, and the evaluation has been overseen by a Reviewing Editor, Pankaj Kapahi, and James Manley as the Senior Editor. The following individual involved in review of your submission have agreed to reveal their identity: Brian M Zid (Reviewer #1).

After consultation with the reviewers, and based on the detailed review below, we have decided that the presented work needs significant revisions to be considered for publication by *eLife* and is thus not suitable for publication in *eLife* in its current form. The Reviewing Editor would be glad to discuss beforehand it makes more sense to consider revision for *eLife* or whether you may want to consider submitting the work elsewhere.

*Reviewer #1 (Recommendations for the authors):*

Do the authors believe that initiation inhibitors don't slow protein production in hsf-1 mutants? If so, this should be shown. Do the authors believe there is a toxic side effect from initiation inhibition in hsf-1 mutants which masks the benefit from reducing protein synthesis in this background? Either way they should discuss this and revise their conclusion for elongation inhibition, similarly to what they have done for initiation inhibition.

One potential alternative explanation for the effect of elongation inhibition in hsf-1 and on proteostasis is that as reduced elongation speed is important for nascent protein folding (Kim et al. Skach Science 2015, Zhou et al. Liu Nature 2013) and coding fidelity (Xie et al. Proud Current Biology 2019), that low doses of elongation inhibitor slow translation elongation speed leading to enhanced proteostasis, outside of just the changes in overall protein synthesis.

*Reviewer #2 (Recommendations for the authors):*

1) Validation of the drugs. Using different compounds to inhibit translation at different states is an interesting approach and having these compounds validated for the *C. elegans* community is very useful. However, the effect of the compounds used in this study should be clearly shown in *C. elegans*, as the whole manuscript relies on the comparison of translation inhibition at selected steps. The effectivity of the drugs to inhibit translation initiation and elongation, respectively, was shown using puromycin incorporation (Figure 1A), which measures changes in overall protein synthesis and not changes in translation rates at selected steps. The analyzed compounds were studied in other systems such as mammalian cells and yeast, however, the source given in this manuscript does not show if they were previously used and validated in *C. elegans*. To make the claim more solid that these drugs do inhibit specific steps of translation in the nematode, the authors should have used additional ways to show how the compounds act on the translation process, such as polysome profiling. For example the proposed translation initiation inhibitor salubrinal is suggested to inhibit eIF2alpha phosphatases (hence inducing ISR and reducing translation initiation). However, to my best knowledge, its mechanism of action is unknown and, more importantly, the eIF2alpha phosphatases of *C. elegans* are not identified, making it speculative if salubrinal does inhibit initiation. Also, while both anisomycin and lycorine are supposed to inhibit elongation and were shown to inhibit overall translation to similar extends (Figure 1A), the strength of their effect on survival during heat shock is different between the drugs (Figure 1D; see also effect of PolyQ aggregation in Sup. Figure S2), raising the question if they indeed act similarly on translation or if they might have off-target effects. For example, anisomycin is discussed to have strong side effects (DOI: 10.1128/MCB.15.9.4930).

2) Consistency of compounds and genotypes used throughout manuscript. Comparing the effects of the inhibition of different translation steps during selected external stresses and in different genetic models is a great approach. Unfortunately, because of the inconsistent use of genotype-treatment-combinations, the lack of Wildtype controls in some assays and splitting graphs based for example on treatment types, it is hard to interpret the results in a global way and to truly compare between initiation and elongation inhibition.

3) Statement that elongation inhibitors reduce ongoing toxicity by reducing folding load independently of hsf-1. Some contradictions in the effect of elongation inhibition are not addressed. Elongation inhibition is discussed to reduce toxicity by reducing protein folding load. While it is shown that inhibition of elongation reduces protein aggregation upon heat shock (Figure 2D,E) and in a PolyQ model (Figure 4C), protein aggregation is not lowered by elongation inhibition in aged worms (Figure 5C,D,E). It is further claimed that elongation inhibition acts independently of hsf-1 in many of the observed phenotypes, but sometimes Wildtype controls are missing to rule out a partial dependency. Furthermore, there could be other factors than hsf-1 involved that are changed upon elongation inhibition and mediate the phenotypes instead of (or in addition to) the overall reduction of protein and hence the increase in folding capacity. Also the inhibition of initiation might reduce folding load (as indicated in Figure 5D,E). More direct data to compare protein aggregation upon inhibition of initiation with elongation during stresses are mainly missing. Translation itself is not compared on a molecular level between the different translation modulation modes.

4) Effects on lifespan. The lifespan assays in Figure 5 are very interesting and underline the previous observations nicely, as they directly compare two initiation and elongation inhibitors in Wildtype and hsp-1 mutant contexts. With regards to elongation inhibition, there might be contradictions to other published data that found inhibition of elongation using cycloheximide to increase lifespan (DOI: 10.1038/srep18722). One more reason why it would be great to have the effect of the elongation inhibitors on translation used here validated.

5) Given the high focus of the manuscript on the difference between the inhibition of translation initiation and elongation, translation itself could have been further analyzed using the different compounds and during selected stresses (for example polysome profiling or ribo seq). The authors claim that their results are consistent with a model proposed by Rogers et al. (2011), saying that lifespan extension through inhibition of translation initiation acts via translation of selected factors. Could the status of some of these factors be checked upon translation inhibitor treatment to verify this?

6) Figure 2D,E; Figure 4C; Figure 5C,D,E pose multiple questions that remain unanswered: What is the difference between age-associated protein misfolding on the one hand and proteotoxicity caused by heat, proteasome dysfunction, and PolyQ stretches on the other hand, and why do animals react differently to it when treated with initiation or elongation inhibitors? Why does inhibition of initiation protect from age-associated protein toxicity but not from proteasome-misfunction-induced protein toxicity? Why does inhibition of elongation protect from proteasome-misfunction-induced protein toxicity but not from age-associated protein aggregation?

*Reviewer #3 (Recommendations for the authors):*

hsf-1's role in EIs but not IIs protecting against proteasomal dysfunction: the data is Figure 3 make this a weak conclusion at best. In the paradigm used, the difference between WT and hsf-1 mutant survival is only 20% and the 'limited protection' vs. 'highly protective effects' of the respective chemical classes are not apparent. From the figure, the impact seems to be similar on both genotypes. The 'morphology' improvements brought about by EIs are not explained and the images in 3G cannot be used to measure 'unc' phenotypes.

In Figures 2B, C, the data for concluding hsf-1 dependence/independence of IIs and EIs thermotolerance is weak. The graphs look moderately different from each other but only one P value is provided (does it apply to both treatments? what is the comparison?) that is not very convincing (P 0.05 vs. 0.02). No details for means/SEM/number of animals and trials are provided either.

Data for lifespan (Figure 5A, B) and thermotolerance (Figures1D,E and 2B, C) survival assays are from one trial only and do not include details (mean, number of animals, SEM). Additional independent trials are critical for verification.

How were the protein quantifications performed for the Puromycin immunohistochemistry (1A) and the insoluble/soluble fractions (2D, E; 5D, E). Were specific regions or bands' intensity measured or the whole lane?

Is the inhibitor treatment for 72h (3 days) or 4 days? The article mentions 72h (eg., line 26) but figures (Figure 1, 2) show 4 days.

---

## [Author Response]

Reviewer #1 (Recommendations for the authors):Do the authors believe that initiation inhibitors don't slow protein production in hsf-1 mutants? If so this should be shown. Do the authors believe there is a toxic side effect from initiation inhibition in hsf-1 mutants which masks the benefit from reducing protein synthesis in this background? Either way they should discuss this and revise their conclusion for elongation inhibition, similarly to what they have done for initiation inhibition.One potential alternative explanation for the effect of elongation inhibition in hsf-1 and on proteostasis is that as reduced elongation speed is important for nascent protein folding (Kim et al. Skach Science 2015, Zhou et al. Liu Nature 2013) and coding fidelity (Xie et al. Proud Current Biology 2019), that low doses of elongation inhibitor slow translation elongation speed leading to enhanced proteostasis, outside of just the changes in overall protein synthesis.

Thank you for this point. As mentioned above and shown in the new Figure 6, initiation inhibitors do not reduce the concentration of newly synthesized proteins in hsf-1(sy441). On the contrary, they surprisingly increase it. Thus, with the evidence that elongation inhibitors protect from proteotoxic stress but do not extend lifespan, it is clear that lowering the concentration of newly synthesized proteins is necessary but not sufficient for lifespan extension.

The idea that translation inhibitors decrease the speed and improve the coding fidelity is a very interesting suggestion. Unfortunately, we do not have the technical ability to measure the speed of translation in vivo in *C. elegans* or to determine the quality of the proteome.

Reviewer #2 (Recommendations for the authors):(1) Validation of the drugs. Using different compounds to inhibit translation at different states is an interesting approach and having these compounds validated for the *C. elegans* community is very useful. However, the effect of the compounds used in this study should be clearly shown in *C. elegans*, as the whole manuscript relies on the comparison of translation inhibition at selected steps. The effectivity of the drugs to inhibit translation initiation and elongation, respectively, was shown using puromycin incorporation (Figure 1A), which measures changes in overall protein synthesis and not changes in translation rates at selected steps. The analyzed compounds were studied in other systems such as mammalian cells and yeast, however, the source given in this manuscript does not show if they were previously used and validated in *C. elegans*. To make the claim more solid that these drugs do inhibit specific steps of translation in the nematode, the authors should have used additional ways to show how the compounds act on the translation process, such as polysome profiling. For example the proposed translation initiation inhibitor salubrinal is suggested to inhibit eIF2alpha phosphatases (hence inducing ISR and reducing translation initiation). However, to my best knowledge, its mechanism of action is unknown and, more importantly, the eIF2alpha phosphatases of C. elegans are not identified, making it speculative if salubrinal does inhibit initiation. Also, while both anisomycin and lycorine are supposed to inhibit elongation and were shown to inhibit overall translation to similar extends (Figure 1A), the strength of their effect on survival during heat shock is different between the drugs (Figure 1D; see also effect of PolyQ aggregation in Sup. Figure S2), raising the question if they indeed act similarly on translation or if they might have off-target effects. For example, anisomycin is discussed to have strong side effects (DOI: 10.1128/MCB.15.9.4930).

These points are all very well taken. One of the aims of this study was to start validating chemical tools to study translation in *C. elegans*. The reviewers' point that we did not validate them far enough was on point. Addressing this point proved far more difficult and was why revising this paper took so long. We started by synthesizing photoaffinity probes that would have allowed us to pull down and identify the *C. elegans* targets of these compounds. However, the chemically modified versions of the inhibitors lost their activity, with only the anisomycin probe still behaving like the original molecule.

Thus we switched strategies and generated the myo-3p::GFP-IRES-tdTomato transgenic worm that allows us to distinguish initiation-dependent from initiation-independent translation. However, as shown in Figure 1F, the variability in fluorescence is large, which caused us difficulties in quantifying the effects precisely. A COPAs bio sorter then allowed us to assay enough animals to unequivocally determine that these inhibitors inhibit translation initiation or elongation as they do in mammals. For 4ER1Cat, we could also show that it no longer extends lifespan in the ifg-1 mutants and that both 4ER1Cat and ifg-1 RNAi surprisingly increase the concentration of newly synthesized proteins in hsf-1 mutants. We thank the reviewer for this criticism, as we think addressing it improved the paper considerably.

(2) Consistency of compounds and genotypes used throughout manuscript. Comparing the effects of the inhibition of different translation steps during selected external stresses and in different genetic models is a great approach. Unfortunately, because of the inconsistent use of genotype-treatment-combinations, the lack of Wildtype controls in some assays and splitting graphs based for example on treatment types, it is hard to interpret the results in a global way and to truly compare between initiation and elongation inhibition.

In some cases, we were not sure what exactly was meant. All the experiments were conducted in parallel, always including N2 +/- treatment control and focusing on the 4 compounds described. We re-arranged some of the figures and hopefully addressed this point. We initially chose to split thermotolerance, proteasome, and survival graphs to make the different responses more obvious. However, these experiments were conducted in parallel. These are clearly noted as Figures 2B, 2C, 4B, 5A 5B.

(3) Statement that elongation inhibitors reduce ongoing toxicity by reducing folding load independently of hsf-1. Some contradictions in the effect of elongation inhibition are not addressed. Elongation inhibition is discussed to reduce toxicity by reducing protein folding load. While it is shown that inhibition of elongation reduces protein aggregation upon heat shock (Figure 2D,E) and in a PolyQ model (Figure 4C), protein aggregation is not lowered by elongation inhibition in aged worms (Figure 5C,D,E). It is further claimed that elongation inhibition acts independently of hsf-1 in many of the observed phenotypes, but sometimes Wildtype controls are missing to rule out a partial dependency. Furthermore, there could be other factors than hsf-1 involved that are changed upon elongation inhibition and mediate the phenotypes instead of (or in addition to) the overall reduction of protein and hence the increase in folding capacity. Also the inhibition of initiation might reduce folding load (as indicated in Figure 5D,E). More direct data to compare protein aggregation upon inhibition of initiation with elongation during stresses are mainly missing. Translation itself is not compared on a molecular level between the different translation modulation modes.

There seem to be several points here. First: It is true that elongation inhibitors could involve other factors. However, the protein folding load model is the simplest version of the model on how elongation inhibitors exert their effect. A key question in this study was to test if we can find conditions where it does not apply. Except for age-associated aggregation, that model correctly predicted all the outcomes for elongation but not initiation inhibitors. For why elongation inhibitors may fail to reduce age-associated protein aggregation please see ANSWER 10 for a potential explanation.

Second, we are unsure where we were missing a wild-type control. The only place we could find was the missing images and length quantification for hsf-1 and Q35. There was no detectable difference, and we thus left it out. Regarding going down to the molecular level, we now show in Figure 1 that these drugs act as initiation or elongation inhibitors and establish a new and unexpected role for HSF1 in Figure 6.

(4) Effects on lifespan. The lifespan assays in Figure 5 are very interesting and underline the previous observations nicely, as they directly compare two initiation and elongation inhibitors in Wildtype and hsp-1 mutant contexts. With regards to elongation inhibition, there might be contradictions to other published data that found inhibition of elongation using cycloheximide to increase lifespan (DOI: 10.1038/srep18722). One more reason why it would be great to have the effect of the elongation inhibitors on translation used here validated.

Thank you for this comment. We were very pleased to find that the inhibitor data, especially the results for 4ER1Cat, agreed so well with the previous results shown by Howard et al. for ifg-1. Such a clear agreement is relatively rare, as genetic or pharmacological inhibition do not necessarily replicate each other. In contrast to knockouts or RNAi, small molecules do not remove the protein but only inhibit a site of a protein. Thus, they should be compared to alleles that affect only one aspect of a protein rather than to knockouts. However, in this case, the pharmacological and genetic results align very well, reciprocally strengthening each other. The hsf-1 dependency of initiation inhibition also aligns very well with Rieckher et al., which we now added to our citation list. The reasons why we did not use cycloheximide and why we think it is problematic to use are outlined in ANSWER 3. As outlined in ANSWER 6, we validated the inhibitor action and appreciate that the reviewer raised this point.

(5) Given the high focus of the manuscript on the difference between the inhibition of translation initiation and elongation, translation itself could have been further analyzed using the different compounds and during selected stresses (for example polysome profiling or ribo seq). The authors claim that their results are consistent with a model proposed by Rogers et al. (2011), saying that lifespan extension through inhibition of translation initiation acts via translation of selected factors. Could the status of some of these factors be checked upon translation inhibitor treatment to verify this?

This is a great suggestion and we will attempt those for at least two of the inhibitors (anisomycin and 4ER1Cat). However, after the reviewer pointed out that a better characterization of the inhibitors, their mode of action, and targets would be beneficial, we first focused on this action and established that 4ER1Cat acts by inhibiting ifg-1.

(6) Figure 2D,E; Figure 4C; Figure 5C,D,E pose multiple questions that remain unanswered: What is the difference between age-associated protein misfolding on the one hand and proteotoxicity caused by heat, proteasome dysfunction, and PolyQ stretches on the other hand, and why do animals react differently to it when treated with initiation or elongation inhibitors? Why does inhibition of initiation protect from age-associated protein toxicity but not from proteasome-misfunction-induced protein toxicity? Why does inhibition of elongation protect from proteasome-misfunction-induced protein toxicity but not from age-associated protein aggregation?

These are excellent questions. At the moment, we only have the observation that this seems to be the case. For example, we were surprised that the elongation inhibitors failed to dampen age-associated protein aggregation but could rescue most other forms or proteotoxic stress. We only have a speculative but probably reasonable answer that is testable. As worms age, their translation rates decline. As a result of diminishing translation, the elongation inhibitor has less of an effect with increasing age. Initiation inhibitors, through selective translation, do not only reduce translation but remodel the entire proteome, most likely leading to a less misfolding-prone proteome. While these ideas are currently speculative, we now have chemical tools to time-dependently inhibit translation to test these ideas. For example, if translation declines with age, the protective effects of elongation inhibitors on any of the proteotoxic stressors we investigated in this study should decline as well. In other words, the protective effect of an elongation inhibitor should be proportional to overall translation and thus decline with age.

Reviewer #3 (Recommendations for the authors):hsf-1's role in EIs but not IIs protecting against proteasomal dysfunction: the data is Figure 3 make this a weak conclusion at best. In the paradigm used, the difference between WT and hsf-1 mutant survival is only 20% and the 'limited protection' vs. 'highly protective effects' of the respective chemical classes are not apparent. From the figure, the impact seems to be similar on both genotypes. The 'morphology' improvements brought about by EIs are not explained and the images in 3G cannot be used to measure 'unc' phenotypes.

There are several aspects to this paragraph. (1) The reviewer is correct in the assessment that the EI improves survival under bortezomib only for hsf-1 and not for N2. Therefore, we have changed the language and now state that the EI provides limited protection in N2.

However, we respectfully disagree concerning hsf-1 and polQ. The effects of the EI on the morphology of bortezomib hsf-1 and polQ animals are striking, especially considering that a translation inhibitor would be expected to make animals smaller. The rescue of survival for hsf-1 is also obvious when doing the experiment and reaches p<0.0001. When measured against the DMSO control of the same genotype in hsf-1(sy441) the elongation inhibitors restore survival almost to the level of the hsf-1(sy441) no-bortezomib control (now Figure 4B).

In Figures 2B, C, the data for concluding hsf-1 dependence/independence of IIs and EIs thermotolerance is weak. The graphs look moderately different from each other but only one P value is provided (does it apply to both treatments? what is the comparison?) that is not very convincing (P 0.05 vs. 0.02). No details for means/SEM/number of animals and trials are provided either.

To clarify this, we have amended Figure 2 and included animal numbers and P-values.

The 0.02 P-value that the reviewer mentions was inserted to show that 4ER1Cat caused a significant increase in survival only at this one-time point in the heat shock assay in the hsf-1(sy441) mutant. While we felt it correct to state that 4ER1Cat is mainly independent of hsf-1 because all other comparisons were not significant, we did not want to ignore this single point. Howard et al. (2016) suggested that the thermotolerance of the ifg-1 mutant (the target of 4ER1Cat) mainly depends on hsf-1 but not wholly. We intended to highlight that our results with a chemical inhibitor match their results with the mutant. In hindsight, this was confusing and made our data look much weaker than they are. In the new version of the figure, we added asterisks to show that the differences are highly significant or not at all. We also specifically state the number of independent experiments included in the figure legends.

Data for lifespan (Figure 5A, B) and thermotolerance (Figures1D,E and 2B, C) survival assays are from one trial only and do not include details (mean, number of animals, SEM). Additional independent trials are critical for verification.

This was our mistake. In the revised version, we now clarify these points. For the lifespan data (Figure 5), we provide an extensive supplementary file containing all the lifespan results with ID numbers referenced in the figure to identify the data set in the table that was used to make the figure. All experiments were repeated between 3 to 7 times. We did not sufficiently state that the graph represents the mean of three independent trials. We added the error bars and the total number of animals (sum of three experiments) indicated in Figure 2 for the heat shock data. We also specifically state the number of independent experiments included in the figure legends.

“Data show the mean ± SEM from three independent trials where each measurement is at least: ** = p ≤ 0.01, *** p ≤ 0.001 and **** p ≤ 0.0001 by row–matched two–way ANOVA with Šídák multiple comparisons test.*”*

How were the protein quantifications performed for the Puromycin immunohistochemistry (1A) and the insoluble/soluble fractions (2D, E; 5D, E). Were specific regions or bands' intensity measured or the whole lane?

We have added the following description to the material and methods to address this concern:

Line 488:

“Gels were quantified in ImageJ by dividing the integrated intensity of each full insoluble lane by the integrated intensity of the corresponding full total protein lane after subtracting a similar area background lane, then normalizing to the DMSO control.”

Line 546:

“To determine the relative intensities of each blot, the integrated intensity (pixel intensity divided by pixel area) was measured for each lane using ImageJ. A similar-sized band with no signal was used to subtract the background then each intensity normalized to its corresponding GAPDH loading control. Finally, each band’s integrated intensity was normalized to the wild-type DMSO control for quantification and statistics. Significance was determined by one–way ANOVA with Šídák multiple comparisons test.”

Is the inhibitor treatment for 72h (3 days) or 4 days? The article mentions 72h (eg., line 26) but figures (Figure 1, 2) show 4 days.

The reviewer is correct. The exposure is from day 1 to day 4, which is 72h or 3 days. We have amended Figure 2. Figure 1 was substantially changed a no longer contains the day 4 reference.